# Olivar: towards automated variant aware primer design for multiplex tiled amplicon sequencing of pathogens

Michael X. Wang [1], Esther G. Lou [2], Nicolae Sapoval [3], Eddie Kim [3], Prashant Kalvapalle [2], Bryce Kille [3], R. A. Leo Elworth [3], Yunxi Liu [3], Yilei Fu [3], Lauren B. Stadler [2] ✉ & Todd J. Treangen [1,3] ✉

Tiled amplicon sequencing has served as an essential tool for tracking the spread and evolution of pathogens. Over 15 million complete SARS-CoV-2 genomes are now publicly available, most sequenced and assembled via tiled amplicon sequencing. While computational tools for tiled amplicon design exist, they require downstream manual optimization both computationally and experimentally, which is slow and costly. Here we present Olivar, a first step towards a fully automated, variant-aware design of tiled amplicons for pathogen genomes. Olivar converts each nucleotide of the target genome into a numeric risk score, capturing undesired sequence features that should be avoided. In a direct comparison with PrimalScheme, we show that Olivar has fewer mismatches overlapping with primers and predicted PCR byproducts. We also compare Olivar head-to-head with ARTIC v4.1, the most widely used primer set for SARS-CoV-2 sequencing, and show Olivar yields similar read mapping rates (~90%) and better coverage to the manually designed ARTIC v4.1 amplicons. We also evaluate Olivar on real wastewater samples and found that Olivar has up to 3-fold higher mapping rates while retaining similar coverage. In summary, Olivar automates and accelerates the generation of tiled amplicons, even in situations of high mutation frequency and/or density. Olivar is available online as a web application at https://olivar.rice.edu and can be installed locally as a command line tool with Bioconda. Source code, installation guide, and usage are available at https://github.com/treangenlab/Olivar.

The devastating COVID-19 pandemic has forever highlighted the utility and importance of biosurveillance for tracking the spread of emerging pathogens. Metagenomic sequencing of environmental samples has enabled the discovery of novel pathogens[1], provided real-time insights into the spread and evolution of infectious disease[2], and enabled the exploration of variant-specific effects on the host[3]. However, the relatively high cost and long turnaround time of metagenomic sequencing remain impractical when a large number of samples and fast turnaround times are necessary, which is the baseline for monitoring pathogens from the environment. Targeted approaches offer advantages in this setting as the ratio of the targeted pathogen is typically low compared to non-target background sequences[4,5] (e.g., monitoring of SARS-CoV-2 in wastewater[6]), making untargeted metagenomic sequencing both more expensive, computationally intensive, and often lacking in sensitivity.

[1]Department of Bioengineering, Rice University, Houston, TX 77030, USA. [2]Department of Civil and Environmental Engineering, Rice University, Houston, TX 77005, USA. [3]Department of Computer Science, Rice University, Houston, TX 77005, USA. ✉e-mail: lauren.stadler@rice.edu; treangen@rice.edu

Thus, targeted amplification or enrichment can both decrease sequencing cost and improve the sequencing sensitivity for pathogen genomes of interest[4,5]. PCR tiling and DNA hybridization probes are two common approaches to amplify whole genomes of specific viruses[5,7]. While DNA hybridization probes are better at preserving the relative abundance of different species, PCR tiling has the advantage of simpler experimental workflow, faster turnaround time, and less DNA input requirement[7]. Therefore, PCR tiling has been widely used to monitor SARS-CoV-2 and characterize SARS-CoV-2 variants[8]. As of December 27th, 2022, there are 14.4 million SARS-CoV-2 genomes on GISAID[9], and 6.5 million available in NCBI GenBank[10], the vast majority of which were sequenced and assembled via tiled amplicon sequencing. However, when combining hundreds of primers within a single tube, PCR tiling has similar pitfalls as multiplexed PCR, including (i) uneven amplification of different genomic regions and (ii) excessive PCR byproducts (e.g., primer dimers and amplification of non-targeted sequences), resulting in a higher cost to reach a minimum acceptable sequencing depth[11]. PCR byproducts can also result in false variant calls[12], requiring manual oversight, re-analysis and slow down the deployment which is critical in the midst of the pandemic. Moreover, PCR primers should be designed to avoid genomic regions with heavy variation[13] (e.g., single-nucleotide polymorphisms or SNPs) and secondary structures[11] to prevent amplicon dropout. Altogether, these pitfalls can lead to higher sequencing cost, uneven coverage, and lower sensitivity, which to date requires one or more runs of experimental validation and manual primer redesign, making the development of a PCR tiling assay costly and labor intensive[14].

Although there are existing tools for designing PCR tiling, some design tiled amplicons as single plex assays[15] or a number of small primer pools (<10 primers)[16], instead of multiplexed assays where tens or hundreds of primers are mixed in the same reaction. Furthermore, previous approaches do not optimize all of the aforementioned criteria simultaneously, nor do they adequately explore the solution space of possible primer combinations[4]. The current state of the art PCR tiling design software tool, PrimalScheme[4], takes a sequential approach to primer design. Specifically, starting from the left side (5' end) of the genome, PrimalScheme sequentially designs each primer until the whole genome is covered, thus, newly designed primers will not affect the choice of previously designed primers. Although PrimalScheme also considers genomic variations, GC content, primer dimers, etc., the choice of new primers will be limited by previously designed primers. For example, the region where a new primer is generated might have high GC content, or candidates of the new primer might form primer dimers with previously designed primers. In the worst-case scenario for PrimalScheme, a gap in tiling will exist since no primer candidate satisfies the design requirements, leading to reduced genomic coverage and/or requiring manual redesign. Thus, its output primers are semi-optimized and often require further tweaking and redesign[14]. This is evidenced by the most widely used PCR tiling primer set in ARTIC[17], initially designed with PrimalScheme and used to sequence millions of SARS-CoV-2 genomes. ARTIC has undergone several iterations of manual tweaking and optimization[13,14,17], while updating the primer set is necessary when new variants of a pathogen emerge, issues such as primer dimer and amplicon dropout could be avoided during in silico design (e.g., the primer dimer issue of ARTIC v4.1[12]).

Here we show Olivar can be used as an end-to-end pipeline for rapid and automatic design of primers for PCR tiling. Olivar accomplishes this by introducing the concept of the "risk" of primer design at the single nucleotide level, enabling fast evaluation of thousands of potential tiled amplicon sets. Olivar looks for designs that avoid regions with high-risk scores based on SNPs, non-specificity, GC contents, and sequence complexity. We select these four components according to known challenges with multiplexed primer design: SNPs represent sequence variation, non-specificity represents the likelihood of non-specific priming, while sequences with extreme GC content and/or low complexity are more likely to be repetitive, bearing secondary structures[18] and producing more primer dimers[11]. Olivar also leverages the SADDLE algorithm[11] to optimize primer dimers in parallel and provides a separate validation module that allows users to evaluate their multiplex PCR primers from various aspects, including the likelihood of dimerization, amplification of non-targeted sequences, etc. To evaluate the performance of our method, we use Olivar to automatically design a set of primers that tile the entire SARS-CoV-2 genome with 146 amplicons in under 30 minutes. In a direct in silico comparison with PrimalScheme, Olivar has lower predicted primer dimerization, fewer SNPs overlapping with primers (4 vs. 18), and fewer predicted non-specific amplifications (5 vs. 27). We conduct an experimental head-to-head comparison with the ARTIC v4.1, the most widely used tiled amplicons for SARS-CoV-2, and find that Olivar has similar mapping rates (~90%) and better coverage for synthetic RNA samples with both low (18) and high (35) cycle threshold (Ct) values. We also test Olivar and ARTIC v4.1 on 4 wastewater samples, and show that Olivar has 1 to 3-fold higher mapping rates and similar coverage. Furthermore, Olivar includes an interactive visualization module that shows the targeted sequence's risk landscape and the primer placement, allowing a convenient overview of the automated design (Fig. 1).

## Results

There are three major steps in Olivar, as illustrated in Fig. 1a,

1. Generation of a risk score for each nucleotide of the targeted sequence based on user-provided inputs (reference sequence, location of SNPs, BLAST database, etc.).
2. Generation and evaluation of primer design region (PDR) sets based on a Loss function.
3. Generation of primer candidates for each PDR and minimization of primer dimers with the SADDLE algorithm.

The primer design region (PDR) is a short DNA sequence (40nt by default) from which primer candidates are generated with SADDLE (Fig. 1b). The risk score emphasizes sequence features that should be avoided for primer design (e.g., high GC regions), and its definition can be tailored for specific applications. Here, we select four components critical to primer design to calculate the risk score: single-nucleotide polymorphisms (SNPs), high/low GC content (extreme GC), homopolymers (low complexity), and repeated sequence across genomes (non-specificity). Each nucleotide is given a score for each of the four sequence features, creating four different arrays named as risk components. Risk components are then weighted with user-defined weights and summed together to generate the risk array (Fig. 1c). The risk array can be visualized as a risk landscape and overlaid with the primers designed, giving users a better understanding of how primers are placed on the targeted sequence (Fig. 1d). The visualization module is included in the Olivar software as well as the Olivar web app. Given an array of risk scores, Olivar can efficiently evaluate thousands of potential PDR sets and choose the best one based on a Loss function. The best PDR set is input to SADDLE and the combination of primer candidates with minimum dimerization likelihood is output as the final primer set. Detailed description of risk components, generation of PDR sets and the Loss function can be found in Methods.

### Optimization of PDRs

The location of PDRs is crucial for primer design since there is little room for optimizing primer candidates if a PDR overlaps with an undesirable sequence feature (e.g., high GC region). The risk array serves as a guidance for the placement of PDRs, enabling rapid and quantitative evaluation of a set of PDRs by a customized Loss function. The above framework is compatible with various applications involving nucleotide sequence design, and it is not restricted to PCR tiling. However, PCR tiling likely takes the most advantage of the risk array

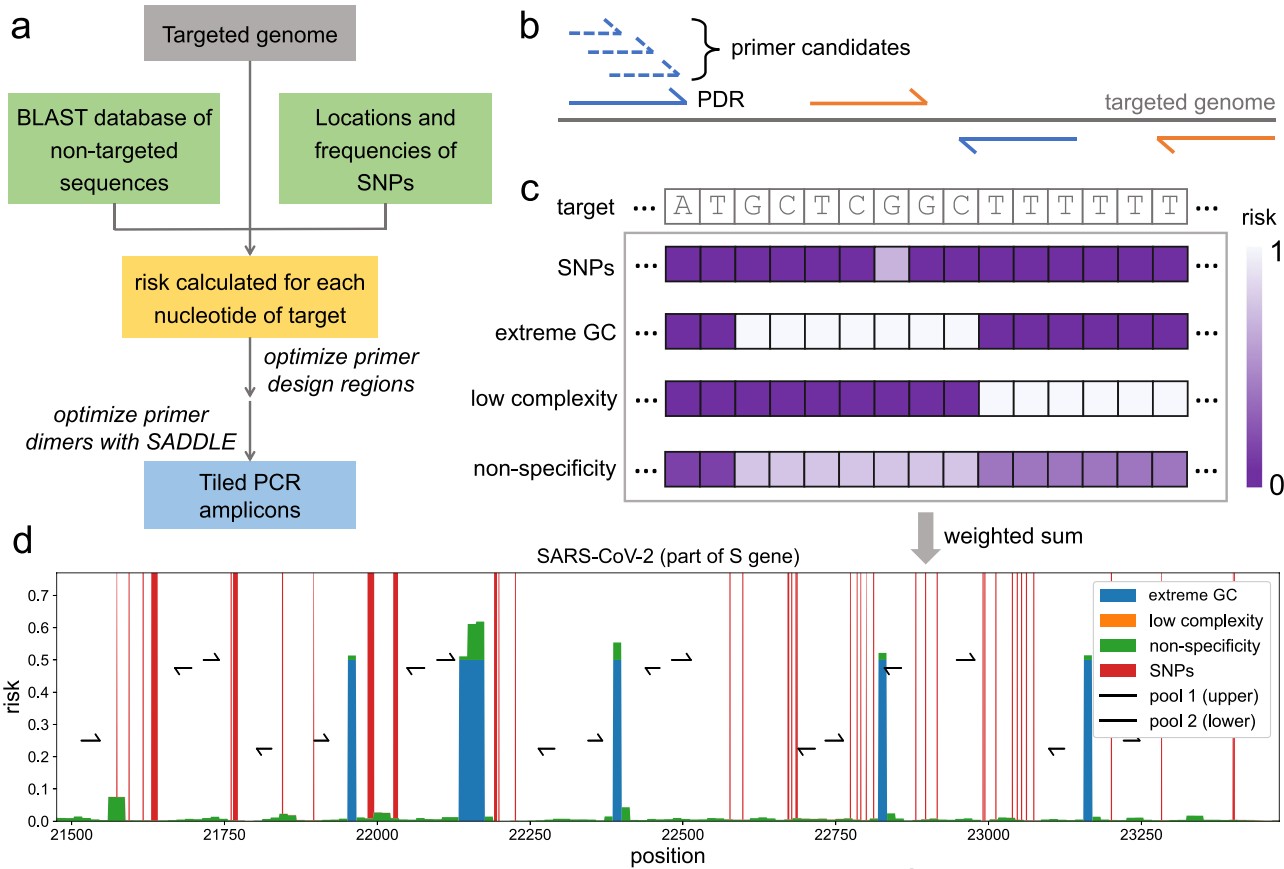

**Fig. 1 | Overall workflow of Olivar and example output. a** The input of Olivar consists of three components: the targeted sequence, a BLAST database built with non-targeted sequences, and single nucleotide polymorphisms (SNPs) to be avoided by primers. Based on the provided inputs, a risk score is calculated for each nucleotide of the targeted sequence. Primer design regions are optimized according to the array of risk scores. One or more primer candidates are generated for each primer design region, and the primer set with minimum primer dimerization is selected with the previously published algorithm SADDLE. **b** A primer design region (PDR) is a short region (40nt by default) on the targeted genome. A pair of PDRs (blue or orange solid lines) covers the genomic region between them, and a valid set of PDRs should cover the whole targeted genome. Primer candidates (blue dashed lines) are generated from each PDR by SADDLE. PDRs are assigned into two pools (pool 1 in blue and pool 2 in orange) to avoid overlapping amplicons. **c** A risk array consists of four components: SNPs, extreme GC content, low sequence complexity and non-specificity, and all of them are calculated for each nucleotide of target and range between 0 and 1. The final risk is calculated as the weighted sum of the four risk components. **d** An example of the risk landscape of the SARS-CoV-2 genome, as well as primers designed by Olivar. The beginning of the S gene is shown, and each risk component is shown in a different color. Different risk components shown in the figure are stacked together instead of overlapping. Primers are assigned into two pools to avoid overlapping amplicons. Together the two pools cover the whole genome.

since there is no explicit requirement for the location of PDRs as long as the targeted sequence is fully covered, leaving plenty of space for optimization. On the contrary, applications such as mutation detection with real-time PCR (qPCR) may require primers to be designed around a certain position of the genome, leaving fewer space for PDR optimization. Other qPCR applications, for example pathogen detection, might not have such restrictions. For a set of PDRs that is "valid" for PCR tiling (here, valid is defined as targeted sequence is fully covered with inserts, where an insert is the sequence between a pair of PDRs), the risk of each PDR as well as the Loss of the PDR set can be readily calculated, allowing thousands of PDR sets to be evaluated within an acceptable amount of time. Specifically, risk of a PDR is the sum of risk scores within it, while the Loss of a PDR set is determined by the risk of the worst PDRs. The definition of the Loss function is based on our experience that the performance of a multiplexed PCR assay is usually undermined by a few "bad players", or primers bearing undesired features such as extreme GC content, low sequence complexity, etc[11]. Instead of calculating the total risk of all PDRs, we focus the Loss function on the worst PDRs to prevent such "bad players". Detailed description about risk of PDRs and the Loss function can be found in the Method section.

Given that PDRs should not overlap with each other, as well as the desired range of amplicon length, a PDR must fall into a certain region of the targeted sequence (green boxes in Fig. 2a). PDRs are generated sequentially and become a set of PDRs so that the whole sequence is covered. Intuitively, one could randomly generate a large number of PDR sets and choose the one with the lowest Loss. However, considering the almost infinite combinations of PDR sets, this approach could be inefficient to find the optimal design. To accelerate the optimization process, we introduced a certain level of greediness into the random generation. Here, the level of greediness means instead of randomly selecting a PDR within that region, high risk PDRs are excluded from the random selection, with high risk defined as PDR risk greater than $X$th percentile. We experimented $X$ from 10 to 90, and as $X$ becomes greater, less greediness is introduced (Fig. 2b, c). Here the reference genome for SARS-CoV-2 is used (GISAID accession: EPI_ISL_402124), with other input data and parameters described in Methods. While Fig. 2b, c indicates that smaller $X$ is better, there is a higher chance of falling into local optima. Hence, we set $X$ as 30 for better universality. Detailed description of generation and optimization of PDR sets can be found in the Methods section. The threshold $X$ is the trade-off between optimization effectiveness and time

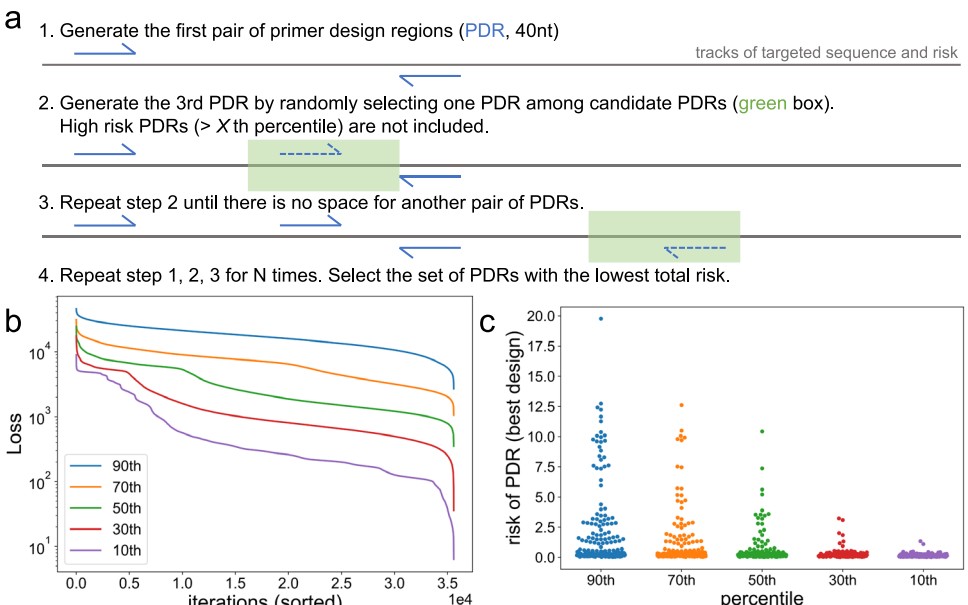

**Fig. 2 | Optimization of primer design regions (PDRs). a** Starting from the 5' end of the targeted sequence, a pair of PDRs is randomly generated. The length of each PDR is fixed to 40nt. Given that PDRs should not overlap with each other, as well as the desired range of amplicon length, a PDR must fall into a certain region of the targeted sequence, as indicated by the green box. The risk of each 40-mer within this region is calculated as the sum of all single nucleotide risk. Low risk 40-mers (≤$X$th percentile) are considered as candidate PDRs and the next PDR is randomly selected among them. PDRs are repeatedly generated until there is no space for another pair of PDRs, and the total risk of the PDR set is calculated. The whole process is repeated N times, with N determined by desired amplicon length and the length of targeted sequence, and the PDR set with the lowest total risk is selected for downstream design. **b, c** Tuning the threshold for determining candidate PDRs, with SARS-CoV-2 genome as the targeted sequence. Detailed description of other design parameters can be found in the methods section. $X$ is set as 30 for following designs. **b** For each percentile threshold, 35,584 PDR sets are generated, and their Loss is sorted. **c** For each percentile threshold, the PDR set with the lowest Loss is selected and the risk of each PDR is shown.

consumption. When $X = 100$, it is guaranteed to find the optimal design, but the time required might be unacceptable. When $X = 0$ (always select the best PDR), it degenerates to a sequential process where no randomness is introduced, with each iteration generating the same set of PDRs.

A risk array can be calculated from a list of SNPs and a BLAST database (Fig. 1), or can be defined in other ways according to the application. Here, we used the entropy data from Nextstrain[19] to demonstrate the versatility of the risk array (Supplementary Data 1). Nextstrain calculates the Shannon's entropy for each base of the SARS-CoV-2 reference genome (GenBank accession: MN908947.3), based on the multiple sequence alignment (MSA) of genomes available on GISAID. Details about entropy calculation can be found in Methods. We ran the Olivar workflow with the Nextstrain entropy data as the risk array, showing Olivar works with different formulations of risk calculation (Supplementary Fig. 1).

**Olivar outperforms PrimalScheme in silico**

We used Olivar and the state-of-the-art PCR tiling design software PrimalScheme (version 1.4.1) to design tiled amplicons for SARS-CoV-2, based on public genomes on GISAID before March 1st, 2022. We first generated SNPs of Delta (B.1.617.2; 3,961,817 genomes) and Omicron (B.1.1.529; 1,258,730 genomes) variants with the software tool Variant Database[20]. 440 SNPs with frequencies greater than 1% were used as input to Olivar and PrimalScheme (Supplementary Data 2). Since PrimalScheme takes no more than 100 input genomes, we created 100 pseudo genomes bearing those 440 SNPs as input to PrimalScheme. We tested six different random assignments of the 440 SNPs, and PrimalScheme output the same design in all six runs, which is expected according to its source code. Human genome assembly GRCh38 was used as the non-targeted BLAST database for Olivar. Desired amplicon length was set to 252nt to 420nt for Olivar and PrimalScheme, with other input parameters kept as default. This setting is different from

the ARTIC primer set[17], with amplicon length ranging from 380nt to 420nt. A wider range of desired amplicon length would increase design flexibility, giving more chance of avoiding SNPs and other unwanted features for both Olivar and PrimalScheme. This could be more beneficial to Olivar since it optimizes primer placement in parallel, taking more advantage of this increased flexibility. Since there is randomness in the Olivar pipeline, specifically in optimizing PDRs, we conducted six runs to test its reproducibility. On the other hand, there is no randomness in the PrimalScheme pipeline and PrimalScheme will generate the same primer set with the same input. We first compared one of the six Olivar designs with the Primalscheme design (Fig. 3a–d). Compared with PrimalScheme, Olivar primers had a lower predicted dimerization likelihood, represented by the dimer score of SADDLE[11] (Fig. 3a, b), as well as lower BLAST hits against human genome (Fig. 3c). The risk score of each primer was also calculated, with Olivar having a lower average risk per primer (0.14 vs. 0.28, Fig. 3d). Across the six Olivar runs, Olivar primers had fewer SNPs overlapping with primers on average (3.67 vs. 18), as well as lower average frequency of those SNPs (3.03% vs. 10.56%) (Fig. 3e). We also predicted the number of non-specific amplicons with BLAST, and Olivar had fewer non-specific amplicons on average compared to PrimalScheme (5 vs. 27, Fig. 3f). In addition, the PrimalScheme design has 10 gaps (regions not covered by amplicon inserts, except for the left and right ends of the genome), with total length of 427nt (Supplementary Fig. 2), while there were no gaps in any of the six Olivar designs. Default parameters of Olivar and PrimalScheme, SNP calling with Variant Database, generation of pseudo genomes for PrimalScheme can be found in "In silico comparison of Olivar and PrimalScheme on Delta and Omicron variants" in Methods. Prediction of non-specific amplicons can be found in "Prediction of non-specific amplicons" Methods. Sequences and coordinates of both the Olivar design and the PrimalScheme design can be found in Supplementary Data 3-6. Details about reproducing the Olivar design can be found in Code Availability.

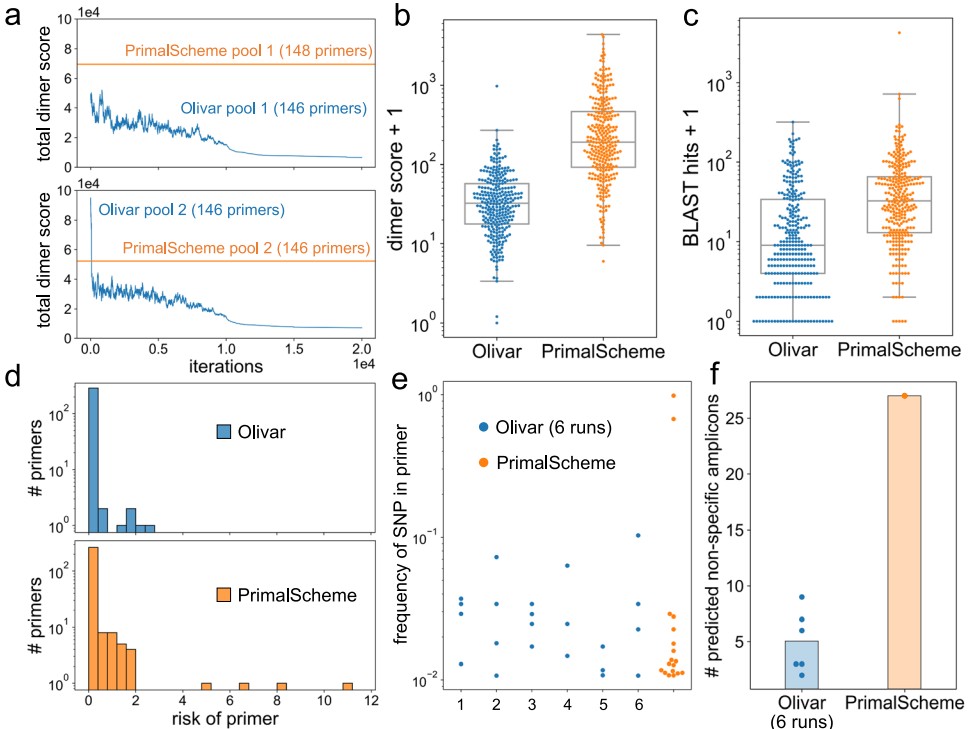

**Fig. 3 | In silico comparison between Olivar and PrimalScheme on a SARS-CoV-2 genome. a** For each primer pool, primer dimerization is optimized with the previously published algorithm SADDLE. SADDLE calculates a dimer score for each primer within a primer pool, as an estimation of primer-primer interaction. Olivar pool 1 has an initial total dimer score of 49,692, and a final score of 6640. Olivar pool 2 has an initial total dimer score of 95,747, and a final score of 7192. PrimalScheme pool 1 and 2 have total dimer scores of 69,474 and 52,369, respectively. **b**, **c** Primers in pool 1 and 2 are shown together, with Olivar having 292 primers and PrimalScheme having 294 primers. Log transformation is performed followed by two-tailed, equal variance t-test. **b** Dimer score is calculated for each primer of Olivar or PrimalScheme ($p = 3.3 \times 10^{-74}$). Box plot elements (Olivar, PrimalScheme): minimum (1, 6.0); lower whisker (3.36, 9.54); first quartile (17.8, 92.6); median (32.3, 191.1); third quartile (57.3, 463.5); upper whisker (270.9, 4387.6); maximum (975.0, 4387.6). **c** The number of hits for each primer is acquired with the BLAST database of non-targeted sequences. Here only human genome is included. The number of

BLAST hits is calculated for each primer of Olivar or PrimalScheme ($p = 1.2 \times 10^{-20}$). Box plot elements (Olivar, PrimalScheme): minimum (1, 1); lower whisker (1, 2); first quartile (4, 13); median (9, 32.5); third quartile (34, 65.7); upper whisker (321, 720); maximum (321, 4268). **d** The risk of a primer is the sum of all risk scores within the primer. Risk distribution is shown for all primers of Olivar or PrimalScheme. Olivar primers have average risk of 0.14, and PrimalScheme primers have average risk of 0.29. **e**, **f** Results of six Olivar runs with the same settings but different random seed. **e** Frequencies of the SNPs that overlap with primers. Out of 440 input SNPs, Olivar primers overlap with 3.67 SNPs on average, with average frequency of 3.03% and highest frequency of 10.72%, while PrimalScheme primers overlap with 18 SNPs with average frequency of 10.56% and highest frequency of 98.31%. **f** Six Olivar designs have 5 predicted non-specific amplicons on average, and the PrimalScheme design has 27 predicted non-specific amplicons. Details about the prediction of non-specific amplicons can be found in Methods.

To further compare the performance of Olivar and PrimalScheme on avoiding SNPs, we chose another set of more diverse genomes: 98 variants of interest (VOI) of SARS-CoV-2. 996 SNPs were called from the MSA of all VOIs, including substitutions, insertions and deletions. We set the desired amplicon length to 380-420, leaving less room for Olivar to optimize primer placement. Olivar had fewer SNPs overlapping with primers in all six runs, as shown in Supplementary Fig. 3. More details can be found in "In silico comparison of Olivar and PrimalScheme on 98 variants of interest (VOI)" in Methods. A list of the 98 VOIs and the 996 SNPs can be found in Supplementary Data 7 and 8.

**Experimental validation of Olivar on synthetic RNA**
To further demonstrate Olivar's performance in real-world applications, we ordered one of the Olivar-designed primer set described above (Fig. 3a–d) and compared Illumina sequencing results with the widely used SARS-CoV-2 primer set ARTIC v4.1. Sequences and coordinates of ARTIC v.4.1 primers can be found in Supplementary Data 9 and 10. Note that ARTIC v4.1 has additional primers added to target the Omicron variants and certain primers have double concentration in the primer pool for better coverage uniformity (Supplementary Data 11), while the Olivar primers were pooled in equal concentration. We first tested both primer sets on synthetic SARS-CoV-2 RNA samples (Twist Bioscience), with different RNA concentrations,

as determined by the Ct value from quantitative real-time PCR (qPCR) targeting the N gene. For both low Ct and high Ct samples, Olivar and ARTIC v4.1 had a similar percentage of sequencing reads mapped to the reference, ranging from 75% to 95%, while Olivar had a lower amount of bases with less than 0.05 × median coverage (Table 1). Detailed description of experimental protocols and analysis of sequencing results can be found in Methods.

**Sequencing SARS-CoV-2 from wastewater with Olivar primers and ARTIC v4.1**
Targeted amplification of pathogen genomes from wastewater samples is challenging since the targeted genomes are highly fragmented, dilute, and comprised of mixtures of circulating variants[6]. We collected 4 wastewater samples from two locations in Houston, USA at two time points. Using the same primer sets and experimental protocol as above, we observed significantly higher mapping rates of Olivar than ARTIC v4.1 ($p = 0.0088$, effect size = 1.27), shown in Table 1. Olivar also had similar percentage of low coverage bases (less than 0.05 × median coverage, $p = 0.39$, effect size = 0.50), compared with ARTIC v4.1 (Table 1). $P$ values are calculated with paired, two-tailed t-test. More details about statistical tests can be found in Methods.

Figure 4a shows the overall genomic coverage of both Olivar and ARTIC v4.1 for one of the wastewater samples (site: CB, Aug. 15, 2022),

**Table 1 | Mapping rate and coverage uniformity of Olivar SARS-CoV-2 primers and ARTIC v4.1 primers**

| | Sample concentration[a] | | Mapping rate[b] | | Less than 0.05 × median coverage[c] | | 0.1 × to 10 × median coverage[c] | |
|---|---|---|---|---|---|---|---|---|
| | | | Olivar | ARTIC | Olivar | ARTIC | Olivar | ARTIC |
| SyntheticRNA control | Ct = 18 (~5 × 10⁵ copy/ul) | replicate 1 | 86.7% | 88.1% | 3.2% | 5.9% | 92.5% | 93.1% |
| | | replicate 2 | 90.2% | 85.9% | 3.7% | 6.0% | 92.7% | 91.5% |
| | Ct = 35 (~3 copy/ul) | replicate 1 | 92.6% | 89.4% | 10.1% | 18.5% | 73.4% | 71.5% |
| | | replicate 2 | 88.8% | 84.5% | 10.7% | 20.6% | 69.8% | 73.3% |
| Wastewatersite: CB | 106.9 copy/ul | Aug. 08, 2022 | 15.0% | 4.6% | 10.4% | 14.1% | 65.6% | 70.5% |
| | 83.9 copy/ul | Aug. 15, 2022 | 52.7% | 42.9% | 8.8% | 15.6% | 77.8% | 71.9% |
| Wastewatersite: KB | 28.1 copy/ul | Aug. 08, 2022 | 22.0% | 10.6% | 19.1% | 17.3% | 64.1% | 59.3% |
| | 35.4 copy/ul | Aug. 15, 2022 | 27.6% | 20.0% | 12.4% | 11.6% | 72.4% | 64.2% |

[a]Concentration of wastewater samples are measured with ddPCR.
[b]Percentage of concordant read pairs mapped to the reference. Average of pool 1 and pool 2.
[c]Percentage of bases with the corresponding coverage. Coverage is the number of reads covering a single nucleotide in the reference genome, normalized with median coverage.

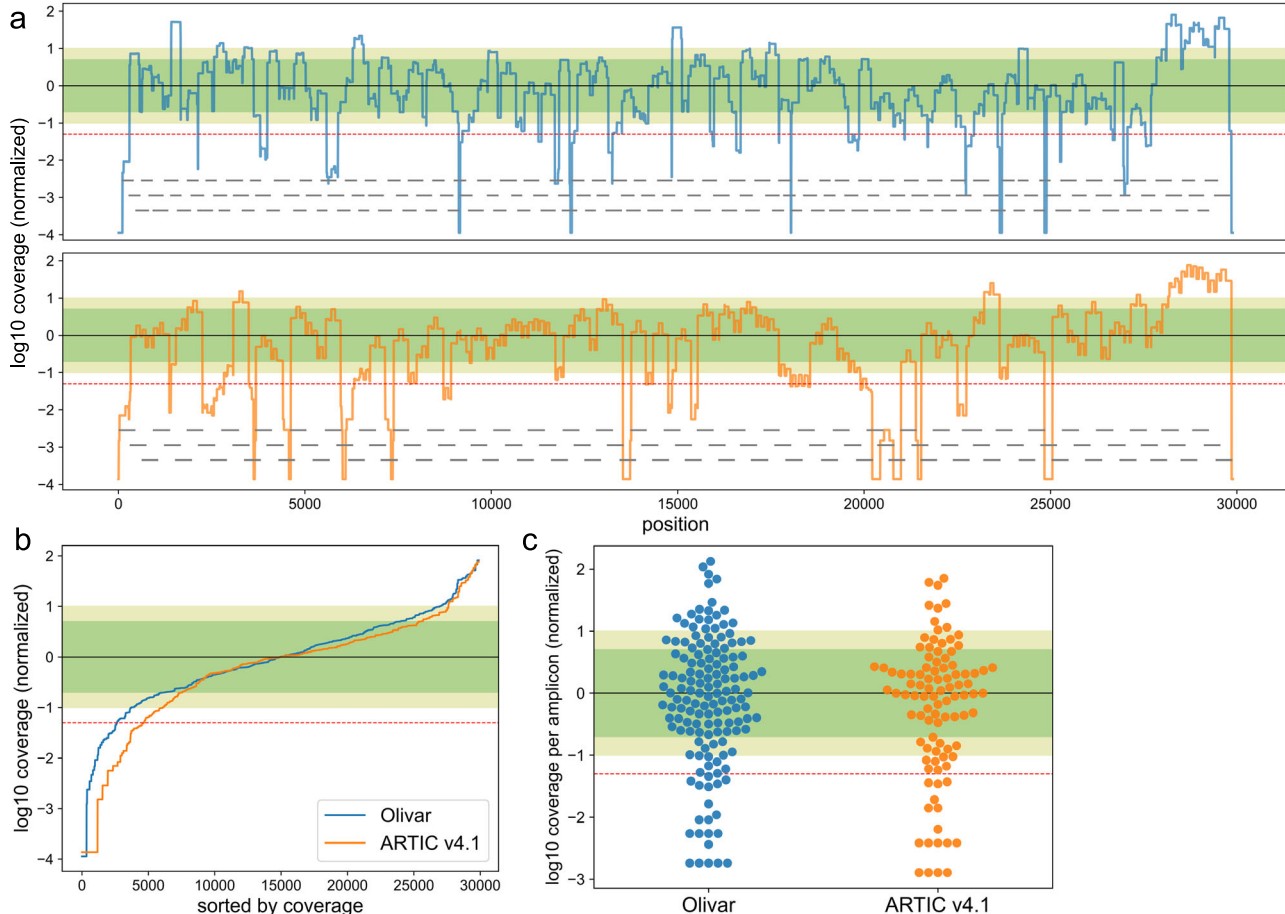

**Fig. 4 | SARS-CoV-2 whole genome coverage of both Olivar (blue) and ARTIC v4.1 (orange) primers.** Figures showing results from one wastewater sample (site: CB, Aug. 15, 2022). Note that Olivar has the highest coverage for this wastewater sample. Please also refer to coverage figures of other samples in Supplementary Information. **a** log10 coverage of each base. Coverage is normalized by median coverage of all bases. Gray lines represent location of amplicons. **b** Sorted log10 coverage of each base. Black solid line represents the median coverage, green shade represents 0.2 × to 5 × median coverage (Olivar: 58.8% bases, ARTIC v4.1: 59.4% bases), olive shade represents 0.1 × to 10 × coverage (Olivar: 77.8% bases, ARTIC v4.1: 71.9% bases), red dashed line represents 0.05 × median coverage (Olivar: 8.8% bases less than 0.05×, ARTIC v4.1: 15.6% bases less than 0.05×). Coverage uniformity of other samples is shown in Table 1. **c** log10 coverage of each amplicon, normalized by median amplicon coverage. (Olivar: 52.7% between 0.2 × to 5 × coverage, 67.8% between 0.1 × to 10 × coverage, 13.7% less than 0.05 × coverage; ARTIC: 52.5% between 0.2 × to 5 × coverage, 67.7% between 0.1 × to 10 × coverage, 16.2% less than 0.05 × coverage).

with amplicon locations shown in gray lines. Note that Olivar has the highest coverage for this sample. To compare the coverage uniformity of Olivar and ARTIC v4.1, genomic locations in Fig. 4a are sorted by coverage (Fig. 4b), showing Olivar and ARTIC v4.1 have 8.8% and 15.6%

of low coverage bases, respectively. Coverage of each amplicon is also shown in Fig. 4c, with Olivar and ARTIC v4.1 having 13.7% and 16.2% low coverage amplicons, respectively. Coverage of other samples can be found in Supplementary Figs. 4–10. More data about mapping rate and

# Olivar

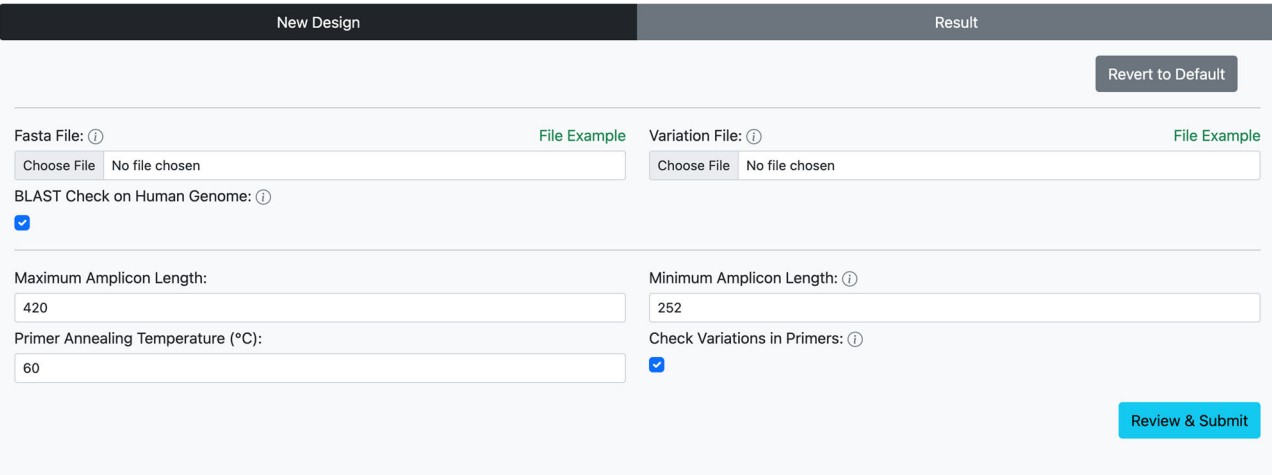

**Fig. 5 | User interface of the Olivar web app.** Users need to upload a reference sequence, and an optional list of SNP locations and frequencies. Other parameters can be adjusted according to application or kept as defaults.

coverage uniformity can be found in Supplementary Data 12 and 13. Details about coverage calculation can be found under "Analysis of sequencing data" in the Methods section.

To make sure the unmapped reads were truly unwanted material (e.g., not from different lineages of SARS-CoV-2), we performed taxonomic assignment with the highly sensitive SeqScreen[21] (version 4.1) with results shown in Supplementary Fig. 11. On average only 0.102% of the unmapped reads were assigned to SARS-CoV-2; thus we conclude those reads were not from other variants of SARS-CoV-2. More details about taxonomy analysis can be found in Methods.

### Olivar web application

To enable easy access for researchers world wide, we developed a user friendly interactive web interface to run Olivar online (Fig. 5). Although it does not support all available functions at the moment, design output is compatible with the local version of Olivar for further analysis. Input files and parameters are the same as the local version of Olivar, with preset defaults parameters and example input files for download. Each run is assigned with a unique identifier for downloading design results at any time.

## Discussion

We have described and presented results on Olivar, an open-source approach designed towards a fully automated, end-to-end computational method for tiled primer design. Olivar quantifies undesired sequence features with single nucleotide risk scores across the whole genome, enabling efficient evaluation and optimization of primer design regions and ultimately improving the performance of output primers. In silico validation shows that Olivar can avoid high-risk regions in the targeted genome, including SNPs, extreme GC content, low sequence complexity, and repetitive sequences. Compared with the state-of-the-art PCR tiling design software PrimalScheme, Olivar has fewer SNPs overlapping with primers as well as fewer predicted primer dimers and non-specific priming. In addition, in a head-to-head comparison with the most commonly used SARS-CoV-2 primer set ARTIC v4.1, Olivar offers equivalent to higher mapping rates and similar genomic coverage on both synthetic RNA samples and wastewater samples. The improvement in mapping rates highlights that Olivar can provide robust designs capable of being produced at lower sequencing costs. While Olivar and ARTIC v4.1 exhibited similar coverage uniformity on wastewater samples, the

ARTIC primer set underwent several versions of optimization, including concentration tuning of primers, whereas the Olivar primers were automatically designed and pooled in equal concentration, saving significant time and cost that comes with multiple rounds of manual redesign.

Olivar is also versatile, not limited to only designing tiled amplicons for viral genomes. For example, PCR tiling is frequently used in applications such as full-length sequencing of entire genes[22]. An Olivar risk array could not only guide the design of tiled amplicons, but also one or a few amplicons when there is no strict requirement on amplicon location (e.g., pathogen detection with digital PCR[6] and measuring copy number variation[23]), by selecting genomic regions with low-risk scores. This could help users quickly find signature sequences ideal for downstream design, such as non-repetitive and highly conserved regions. Furthermore, the modular nature of calculating a risk array allows the customization of risk components. For example, in applications where SNPs are needed to distinguish similar species or strains of pathogens[24], users could define risk scores for sensitivity and specificity to find sequences targeting desired strains while avoiding unwanted ones.

Despite the focus towards an automated design workflow, user's input of sequence variation information and a BLAST database of background non-specific sequences are needed to describe the design problem sufficiently. For sequence variation, it is represented with a list of locations and frequencies and input to Olivar. Another way to represent sequence variation is directly using the group of genomic sequences to be covered (e.g., strains of a certain virus)[4]. While the latter is usually more readily available than the former for viruses and bacteria, it needs more computational resources to evaluate the sensitivity of a PDR for a group of genomes through local alignment or multiple sequence alignment (MSA), especially when the number of genomes is large. For well-studied pathogens, such as SARS-CoV-2 and human monkeypox virus, MSAs are updated in real-time by public databases such as GISAID[9], and coordinates and frequencies of nucleotide change are available at Nextstrain[19]. For species without publicly available MSAs, tools such as Parsnp[25] can efficiently build core-genome alignment and make variant calls.

Another user-defined input for Olivar is the BLAST database for background non-targeted sequences. While we provide the BLAST database for the human genome as it is frequently considered background, users will likely need more comprehensive and

application-specific databases to further reduce non-specific byproducts. Leveraging GC content and sequence complexity can also help avoid low-complexity sequences. We are actively building more background databases for various scenarios, such as pathogen detection in wastewater. Additional background sequence screening improvements will be included in future updates of Olivar.

Except for user input data, parameters in the design workflow could also affect the output primer set. The parameters in Olivar workflow can be categorized into two groups: 1) parameters in risk array calculation, and 2) parameters in the optimization phase, including PDR optimization and primer dimer optimization. The first group of parameters are set based on the fact that most PCR tiling assays are run under annealing temperature of 50–70 °C, with 55–65 °C being the most frequent setting. That results in primer length of 18nt-35nt, and we determined PDR length of 40nt to ensure there are enough primer candidates, while limiting the uncertainty of the final position of the primer. The generation of the risk array and its parameters are then determined based on the length of PDR (e.g, word size of 28, sequence complexity threshold of 0.4, and BLAST parameters, etc., with details described in the Method section). We also introduce weights for different risk components to allow more flexibility for users with expertise and have specific requirements for their design. Within the second group of parameters, parameters related to primer dimer optimization are well established and are adjusted automatically based on the size of the genome[11]. For PDR optimization, we investigated the level of greediness introduced in selecting PDR candidates and chose the optimal parameter based on in silico experiments, as shown in Fig. 2.

To our knowledge, Olivar is the first open-source computational tool designed towards enabling fully automated design of multiplexed PCR tiling assays while considering SNPs, primer dimers, and non-specific amplification simultaneously, with the potential of significantly reducing manual redesign while maintaining low cost. We anticipate that Olivar will prove useful in the surveillance of future infectious disease outbreaks by providing an open-source tool for designing tiled amplicons.

## Methods

### Generation of risk array

A risk array is an array of non-negative real numbers

$$\mathbf{r} = [r_1, r_2, r_3, \ldots, r_L] \tag{1}$$

where $L$ is the length of the targeted sequence. The array $\mathbf{r}$ consists of four weighted components,

$$\mathbf{r} = a_1 \cdot \mathbf{snp} + a_2 \cdot \mathbf{egc} + a_3 \cdot \mathbf{lc} + a_4 \cdot \mathbf{ns} \tag{2}$$

where $a_1$, $a_2$, $a_3$ and $a_4$ are set to 1 as default. $\mathbf{snp}$ represent SNPs, $\mathbf{egc}$ represent extreme GC content, $\mathbf{lc}$ represent low sequence complexity and $\mathbf{ns}$ represent non-specificity. For each user provided single nucleotide polymorphism (SNP) at position p of the targeted sequence,

$$\mathbf{snp}[p] = 10\sqrt{\mathrm{freq}} \tag{3}$$

where freq could be the frequency of the SNP to be avoided. The root of SNP frequency is taken to amplify low frequency SNPs, and timed by a factor of 10 to be comparable to other risk components. Note that freq could also be user defined (e.g., when certain SNPs are considered more important than others, or simply just scale up all frequencies by a certain factor). To calculate the other three components, a set of equal-length, overlapping, and evenly distributed "words" are generated from the targeted sequence. Suppose the length of each word is ws (28 by default), $L$ is divisible by ws and ws is divisible by a positive integer $c$

(2 by default), then the distance between neighboring words is ws/$c$, and

$$\begin{aligned}
\mathbf{W} = \{ & \mathbf{seq}[1 : \mathrm{ws}], \\
& \mathbf{seq}\left[1 + \frac{\mathrm{ws}}{c} : \mathrm{ws} + \frac{\mathrm{ws}}{c}\right], \ldots, \\
& \mathbf{seq}\left[1 + i \cdot \frac{\mathrm{ws}}{c} : \mathrm{ws} + i \cdot \frac{\mathrm{ws}}{c}\right], \ldots, \\
& \mathbf{seq}\left[L - \mathrm{ws} + 1 : L\right]\}
\end{aligned} \tag{4}$$

where $\mathbf{W}$ is the set of words, $\mathbf{seq}$ is the targeted sequence and $i$ is a positive integer. If $L$ is not divisible by ws, the targeted sequence is minimally trimmed to satisfy this condition, and corresponding elements in risk array $\mathbf{r}$ are set to 0. The value of ws is set as 28 by default based on typical primer length (20-25bp) and the minimum word size (7) of the NCBI BLAST+ program[26]. $c$ is set as 2 by default so that each base is covered by two words. GC content and sequence complexity of each word is calculated. Sequence complexity is calculated based on Shannon entropy (described below). The number of BLAST hits of each word is acquired with the user-provided BLAST database. The following parameters are set to the NCBI BLAST+ program[26] (version 2.12.0): evalue as 5, reward as 1, penalty as −3, gapopen as 5, gapextend as 2. These are the default parameters of the task "blastn-short". For a nucleotide at position $p$ ($\mathrm{ws} - \frac{\mathrm{ws}}{c} + 1 \leq p \leq L - \mathrm{ws} + \frac{\mathrm{ws}}{c}$) of the targeted sequence (except for head and tail regions), there is a set of $c$ words overlapping with that nucleotide, denoted as $\mathbf{W_p} \subset \mathbf{W}$. The average GC content of $\mathbf{W_p}$, the average sequence complexity of $\mathbf{W_p}$ and the average number of BLAST hits of $\mathbf{W_p}$ are denoted as gc, cmplx and hits, respectively. $\mathbf{nsraw}$ is an array of 0s with length of $L$. Then,

$$\mathbf{egc}[p] = \begin{cases} 0 & (\mathrm{gc}_{\min} < \mathrm{gc} < \mathrm{gc}_{\max}) \\ 1 & (\mathrm{gc} \leq \mathrm{gc}_{\min} \text{ or } \mathrm{gc} \geq \mathrm{gc}_{\max}) \end{cases} \tag{5}$$

$$\mathbf{lc}[p] = \begin{cases} 0 & (\mathrm{cmplx} > \mathrm{cmplx}_{\mathrm{low}}) \\ 1 & (\mathrm{cmplx} \leq \mathrm{cmplx}_{\mathrm{low}}) \end{cases} \tag{6}$$

$$\mathbf{nsraw}[p] = \mathrm{hits} \tag{7}$$

where $\mathrm{gc}_{\min}$ (0.25 by default), $\mathrm{gc}_{\max}$ (0.75 by default) and $\mathrm{cmplx}_{\mathrm{low}}$ (0.4 by default) are user defined. $\mathbf{nsraw}$ is then normalized to get $\mathbf{ns}$

$$\mathbf{ns} = \frac{\mathbf{nsraw}}{\max(\mathbf{nsraw})} \tag{8}$$

**Calculation of sequence complexity.** Sequence complexity is calculated based on Shannon's entropy. For a DNA sequence $\mathbf{S}$ of length $L$ ($L > 3$) with alphabet {A, T, C, G}, the number of each $k$-mer ($k = 1, 2, 3$) is stored as an array

$$\mathbf{c_k} = [m_0, m_1, \ldots, m_n] \tag{9}$$

where $m_i (i = 0, 1, \ldots, n)$ is the number of a certain k-mer, and $n = 4^k$. For example, if $\mathbf{S} = \mathrm{ACGCAGCGAGCAG}$, then $\mathbf{c_1} = [4, 4, 5]$ since there are 4 'A's, 4 'C's and 5 'G's. Then,

$$\mathbf{p_k} = \frac{\mathbf{c_k}}{L - k + 1} \tag{10}$$

$$e_k = -\frac{\mathbf{p_k} \cdot \log_2(\mathbf{p_k})}{2k} \tag{11}$$

where $e_k$ is the dot product of the two vectors. The complexity of $\mathbf{S}$ is the smallest of $e_1$, $e_2$ and $e_3$.

## Optimization of PDRs

PDRs are regions where primer candidates are generated, and primer candidates are subsequences of a PDR. A PDR is defined by its start coordinate $p$ and stop coordinate $p + l - 1$ (closed interval), where $l$ is the length of the PDR (40 by default). All PDRs have the same length. The risk of the PDR is then defined as sum($\mathbf{r}[p : p + l - 1]$), where $\mathbf{r}$ is the risk array.

**Generation of one PDR.** Each PDR is randomly selected within a certain region $\mathbf{C}$ (Supplementary Fig. 12, also see green boxes in Fig. 2a), the length of $\mathbf{C}$ is greater than or equal to $l$. The risk of all possible PDRs within $\mathbf{C}$ is calculated, and PDRs with risk below $X$th percentile are considered candidate PDRs. Here percentile is defined as a PDR risk threshold below which percentage $X$ of PDRs fall. In short, not all possible PDRs within region $\mathbf{C}$ is included for random selection, and the worst PDRs are excluded to accelerate the optimization process. Therefore, a lower $X$ means a more stringent selection on PDR candidates. $X$ is set to 30 by default to balance optimization efficiency and effectiveness (Fig. 2). This is also discussed in "Optimization of PDRs" in Results.

**Generation of a set of PDRs.** Similar to forward primer (fP) and reverse primer (rP), there are also forward PDR (fPDR) and reverse PDR (rPDR). Here forward means closer to the 5′ end (left end) of the targeted sequence. There are three restrictions for a valid PDR set given a targeted sequence:

1. PDRs should not overlap with each other (this guarantees that primers will not overlap).
2. All amplicons generated from a PDR pair should satisfy a range of desired amplicon length, as defined by the user. An amplicon is defined as a pair of fP and rP, and its length is defined from the start of fP to the end of rP.
3. The targeted sequence are fully covered with inserts. An insert is the region between a PDR pair. Suppose a valid PDR set consists of $N$ PDR pairs and each PDR pair is generated sequentially. For the $k$th PDR pair ($k \geq 1$), the fPDR is selected from region $\mathbf{C_{2k-1}}$, and the coordinates of the fPDR is $[p_{2k-1}, p_{2k-1} + l - 1]$. The rPDR is selected from region $\mathbf{C_{2k}}$, and the coordinates of the rPDR is $[p_{2k}, p_{2k} + l - 1]$. $\text{ALEN}_{\min}$ is minimum amplicon length and $\text{ALEN}_{\max}$ is maximum amplicon length.

Generation of a PDR set starts with the first PDR pair ($k = 1$). Starting from the left end of the risk array and the targeted sequence, the fPDR of the first PDR pair is selected from region $\mathbf{C_1}$ (Supplementary Fig. 12),

$$\mathbf{C_1} = [1, 3l) \tag{12}$$

Here $\mathbf{C_1}$ starts at 1 for simplicity, and region size of $3l$ gives enough room for selecting the first PDR. There are $2l + 1$ possible PDRs within $\mathbf{C_1}$, and one of them is selected as the first fPDR, with start coordinate $p_1$, based on the method described in the section above (Generation of one PDR). The first rPDR is selected from region $\mathbf{C_2}$ (Supplementary Fig. 13), and based on the amplicon length restriction,

$$\mathbf{C_2} = [p_1 + \text{ALEN}_{\min} - l, p_1 + \text{ALEN}_{\max} - 1] \tag{13}$$

The start coordinate of the rPDR is selected as $p_2$.

For the second PDR pair ($k = 2$), the fPDR and rPDR are selected from region $\mathbf{C_3}$ and $\mathbf{C_4}$, respectively. Considering the three restrictions, the boundaries of $\mathbf{C_3}$ are

$$\mathbf{C_3} = [\max(p_1 + l, p_2 + 3l - \text{ALEN}_{\max}), \\ p_2 - 1] \tag{14}$$

Also see Supplementary Fig. 14. The second fPDR is selected from $\mathbf{C_3}$ with start coordinate $p_3$.

The boundaries of $\mathbf{C_4}$ are

$$\mathbf{C_4} = [\max(p_3 - l + \text{ALEN}_{\min}, p_2 + 2l), \\ p_3 + \text{ALEN}_{\max} - 1] \tag{15}$$

Also see Supplementary Fig. 15. The second rPDR is selected from $\mathbf{C_4}$ with start coordinate $p_4$.

Starting from the third PDR pair ($k \geq 3$), the fPDR and rPDR are selected from region $\mathbf{C_{2k-1}}$ and $\mathbf{C_{2k}}$, respectively. Considering the three restrictions, the boundaries of $\mathbf{C_{2k-1}}$ are

$$\mathbf{C_{2k-1}} = [\max(p_{2k-4} + l, p_{2k-2} + 3l - \text{ALEN}_{\max}), \\ p_{2k-2} - 1] \tag{16}$$

Also see Supplementary Fig. 16. The $k$th fPDR is selected from $\mathbf{C_{2k-1}}$ with start coordinate $p_{2k-1}$.

The boundaries of $\mathbf{C_{2k}}$ are

$$\mathbf{C_{2k}} = [\max(p_{2k-1} - l + \text{ALEN}_{\min}, p_{2k-2} + 2l), \\ p_{2k-1} + \text{ALEN}_{\max} - 1] \tag{17}$$

Also see Supplementary Fig. 17. The $k$th rPDR is selected from $\mathbf{C_{2k}}$ with start coordinate $p_{2k}$.

PDR generation will stop when $\mathbf{C_{2k}}$ exceeds the limit of the targeted sequence.

**Loss of a PDR set.** The Loss of a PDR set is defined as the squared sum (sum of squares) of the risk of the top 10% high-risk PDRs. Instead of calculating the total risk of all PDRs, we focus the Loss function on high-risk PDRs and amplify the contribution of the worst PDRs with non-linear transformation (e.g., sum of squares). For an input genome of length $L$, the number of PDR sets generated is $\lfloor 500L/\text{ALEN}_{\max} \rfloor$. The PDR set with the lowest Loss is selected for downstream design.

**Primer pool assignment.** If the targeted sequence is fully covered with inserts (region between fP and rP), then amplicons must be overlapping and the $k$th fP could form a short amplicon with the $(k-1)$th rP. These short amplicons are usually much shorter than desired amplicons, having much higher amplification efficiency. To avoid the formation of these short amplicons, most PCR tiling applications assign amplicons into two pools so that amplicons within each pool are non-overlapping. Suppose the first PDR pair is assigned to pool 1 and the second PDR pair is assigned to pool 2, then the $(2k-1)$th PDR pair is assigned to pool 1 and the $2k$th PDR pair is assigned to pool 2 ($k \geq 2$).

## Optimizing primer candidates with SADDLE

In the previous step the optimal PDR set is separated into two pools of PDRs, and each pool is input to SADDLE separately. Primer candidates are first generated for each PDR, with default parameters: temperature as 60 °C, salinity as 0.18, max primer length as 36, max $\Delta G$ (Gibbs free energy) as −11.8. One primer candidate is randomly picked from each PDR to form a primer set. SADDLE minimize primer dimers within the randomly generated primer set by iteratively replacing a primer candidate with another one from the same PDR. The number of iterations is determined by the total number of PDRs in the pool. The final optimized primer set will be the output of Olivar.

## Prediction of non-specific amplicons

The prediction of non-specific amplicons starts with running BLAST through each single primer, with the following parameters set to the NCBI BLAST+ program[26]: evalue as 5000, reward as 1, penalty as −1, gapopen as 2, gapextend as 1. The values of these parameters are determined based on Primer BLAST[27]. For each BLAST hit, the BLAST program outputs the location and orientation of that hit. A single primer could have multiple BLAST hits, distributed to different

chromosomes. For each chromosome, the location and orientation of hits of all primers are analyzed together. If two hits are close enough and the orientations are legitimate for PCR, that pair of hits is reported as a predicted non-specific amplicon.

### Calculation of Nextstrain nucleotide entropy

Nextstrain[19] calculates Shannon's entropy for each nucleotide position of the reference SARS-CoV-2 genome based on MSA of genomes available on GISAID[9]. Here we include global genomes dating from Dec. 21, 2019 to Dec. 04, 2022. Entropy at a given position $c$ is calculated as below. Suppose we have $N$ genomes. At position $c$, $N_1$ genomes has 'A', $N_2$ genomes has 'T', $N_3$ genomes has 'C', $N_4$ genomes has 'G'. Suppose $N_1$, $N_2$, $N_3$ and $N_4$ are non-zero.

$$\mathbf{p_c} = \left[ \frac{N_1}{N}, \frac{N_2}{N}, \frac{N_3}{N}, \frac{N_4}{N} \right] \tag{18}$$

$$e_c = - \mathbf{p_c} \cdot \ln(\mathbf{p_c}) \tag{19}$$

where $p_c$ is an array and $e_c$ is dot product and entropy at position $c$. Of 7774 locations with non-zero entropy, we selected 1517 locations with entropy greater than 0.01 and input to Olivar as SNP locations and frequencies. A list of locations and their corresponding entropy can be found in Supplementary Data 1. We used the reference genome provided by Nextstrain (GenBank MN908947.3) without a BLAST database of non-targeted sequences. Other parameters are kept as default.

### In silico comparison of Olivar and PrimalScheme on Delta and Omicron variants

For Olivar, we used GISAID genome EPI_ISL_402124 as reference, a BLAST database of human genome assembly GRCh38 as non-targeted sequences, and 440 SNPs of SARS-CoV-2 lineage B.1.617.2 and B.1.1.529 generated with Variant Database (described below). BLAST version 2.12.0 was used to create the BLAST database, with 24 chromosomes and mitochondria of GRCh38 as input sequences. The same BLAST version was used by Olivar to fetch non-specific hits. A different version of BLAST may generate different non-specific results and lead to different Olivar designs. Desired amplicon length was set to 252nt to 420 nt and the --check-var option of was turned on. For the six Olivar runs, random seed was set to 10, 11, 12, 13, 14, 15, respectively. Other Olivar parameters were kept as default. Olivar design of random seed set as 10 was used for comparison with ARTIC v4.1 (Supplementary Data 3 and 4). Olivar version≤1.1.5 should be used to reproduce the results.

For PrimalScheme (version 1.4.1), desired amplicon length was also set to 252nt to 420 nt and 100 artificial genomes were used as input (described below). Other parameters were kept as default.

### SNP calling of Delta and Omicron variants with Variant Database.

We downloaded MSA and metadata of GISAID SARS-CoV-2 genomes before Feb. 28, 2022 and used Variant Database[20] (version 2.4) to output SNPs of lineage B.1.617.2 (Delta variant, 3,961,817 genomes) and B.1.1.529 (Omicron variant, 1,258,730 genomes), with GISAID genome EPI_ISL_402124 as reference for coordinates. Variant Database compares the differences between reference genome EPI_ISL_402124 and all genomes in lineage B.1.617.2 (or B.1.1.529), and SNP frequency is defined as the percentage of genomes that are different from the reference at a given location. Note that Variant Database version 2.4 does not output insertions, so here SNPs include substitutions and deletions. We used the command "frequencies <cluster>" to list the frequencies of individual mutations for B.1.617.2 and B.1.1.529 separately. Insertions are not generated due to the limitations of the Variant Database. Of the 10,000 output SNPs, we kept 440 SNPs with a

frequency greater than 0.01 and input to Olivar. A list of those SNPs can be found in Supplementary Data 2.

**Input genomes for PrimalScheme.** PrimalScheme limits the number of input genomes to 100. Since more than 5 million Delta and Omicron genomes are available from GISAID, 99 artificial genomes are created, bearing the same 440 SNPs input to Olivar. Each SNP is then randomly added to $n$ of the 99 copies of the GISAID reference EPI_ISL_402124,

$$n = \begin{cases} \lfloor 100f \rfloor & (f < 0.5) \\ \lfloor 25 + 50f \rfloor & (0.5 < f < 1) \end{cases} \tag{20}$$

where $f$ is the frequency of the SNP and $\lfloor \rfloor$ means rounding down. Together with the reference, 100 genomes was input to PrimalScheme. We did not keep the original frequency for high-frequency SNPs in the artificial genomes because PrimalScheme will consider a position to be conserved if the vast majority of genomes have the same SNP and that position will not be avoided.

### In silico comparison of Olivar and PrimalScheme on 98 variants of interest (VOI)

MSA of 98 VOIs was directly downloaded from GISAID on October 23, 2023. 996 SNPs are called from the MSA of all VOIs, including substitutions, insertions and deletions, with GISAID genome EPI_ISL_402124 used as reference for coordinates. For Olivar input, we used GISAID genome EPI_ISL_402124 as reference, as well as the 996 SNPs. No BLAST database was used. The 98 VOIs were directly used as input for PrimalScheme. For Olivar and PrimalScheme (version 1.4.1), desired amplicon length is set to 380nt to 420 nt. Olivar was also run six times, with random seed set to 10, 11, 12, 13, 14, 15, respectively. Other Olivar and PrimalScheme parameters are kept as default. A list of the 98 VOIs and the 996 SNPs can be found in Supplementary Data 7 and 8.

### Sample preparation and quantification

Synthetic RNA control for SARS-CoV-2 is purchased from Twist Bioscience (part number 105204, GISAID accession: EPI_ISL_6841980, GISAID name: Hong Kong/HKU-211129-001/2021, B.1.1.529/BA.1, Omicron variant). Time-weighted composite samples of raw wastewater were collected every 1h for 24 h from the influent of the two domestic wastewater treatment plants (WWTPs), CB and KB, on two separate dates (Aug. 8 and Aug. 15, 2022). Samples were kept on ice during transport and stored at 4 °C in the laboratory to be processed within 24 h of collection. Sample concentration using HA filtration and bead beating methods, as well as RNA extraction method using the Chemagic™ Prime Viral DNA/RNA 300 Kit H96 (Chemagic, CMG-1433, PerkinElmer) were as decribed by Lou et al.[6]. To normalize synthetic RNA control to Ct values of 18 and 35, One-step RT-qPCR were performed with qPCRBIO probe 1-Step Go LoROX (PB25.41, PCR Biosystems) on the QuantStudio 3 Real Time PCR System (A28567, Applied Biosystems) as previously described by Lou et al.[28]. These two Ct bounds were selected to represent the high and low concentrations of SARS-CoV-2. SARS-CoV-2 concentrations of wastewater samples were quantified using RT-ddPCR on a QX200 AutoDG Droplet Digital PCR System (Bio-Rad) and a C1000 Thermal Cycler (Bio-Rad) in 96-well optical plates, as previously described by Lou et al.[6].

### Multiplexed PCR, library preparation and sequencing

ARTIC v4.1 primer panel was purchased from Integrated DNA Technologies (IDT, Artic v4.1 NCOV-2019 Panel, 500rxn, 10011442). Primer pool was prepared by IDT according to the instructions provided by the ARTIC network. Specifically, additional primers were added and certain primers were pooled with double concentration. A copy of the instructions and the link to the original web page can be found in Supplementary Data 11. Olivar primers (Supplementary Data 3 and 4)

were ordered in tubes from Sigma Aldrich and mixed by hand to achieve the final equal concentration of 15 nanomolar (nM) per primer. Reverse transcription of synthetic RNA control and extracted waste-water RNA were conducted using 8 uL of sample RNA and LunaScript RT SuperMix kit (NEB, E3010), as described by Tyson et al.[17]. To avoid bias attributed to reverse transcription, for each sample, the total volume of 10 uL cDNA product were gently homogenized by pipetting then divided into four 2.5 uL aliquots for the downstream PCR ampli-fication reactions (using primer pool 1 and 2 of ARTIC v4.1, and using primer pool 1 and 2 of Olivar). PCR amplification was also performed using Q5 Hot Start High-Fidelity 2X Master Mix (NEB M0494) as described by Tyson et al.[17]. PCR products were purified using AmPur-eTM XP beads (Beckman Coulter Inc., A63880). A high bead-to-sample ratio of 1.8 was applied to maximize the potentials of capturing PCR byproduct. Purified DNA samples were normalized to 20 ng/uL in 25 uL and submitted for amplicon sequencing service at Azenta (EZ-Ampli-con), with paired-end (2 × 250bp), quality filtered and adapters trim-med Illumina reads output. The concentration of amplicons was measured using Qubit dsDNA HS kit and a Qubit 2.0 fluorometer (Invitrogen).

### Analysis of sequencing data

Paired-end sequencing reads were mapped to the reference genome (GISAID ID EPI_ISL_402124) with Bowtie2 (version 2.4.5)[29], with max-imum fragment length for valid paired-end alignments (-X) set to 1000. Only read pairs that were mapped concordantly were used for down-stream analysis. Discordant, single-aligned and unmapped read pairs were discarded. Coverage of each reference position was calcu-lated as the number of reads overlapping that position, using PySAM (version 0.19.1)[30], and then normalized with median coverage. Note that reads from the two primer pools were merged before calculating coverage. Sequencing reads were also mapped to amplicon sequences with Bowtie2, and the coverage for each amplicon was the number of read pairs mapped to that amplicon. Data about mapping rate and coverage uniformity can be found in Supplementary Data 12 and 13.

**Statistical tests.** For mapping rates, the two primer pools were treated as different samples ($n = 8$). For coverage uniformity, pool 1 and pool 2 were combined ($n = 4$), and the percentage of genomic positions with less than $0.05 \times$ median coverage was used as a measurement of cov-erage uniformity. P values were calculated with a paired, two-tailed t-test. Effect sizes were calculated as Cohen's d, with the formula below,

$$d = \left| \frac{\overline{D}}{SD_D} \right| \qquad (21)$$

where $d$ is Cohen's d, $\overline{D}$ is the mean of paired difference, and $SD_D$ is the standard deviation of paired difference.

### Taxonomic assignment of unmapped reads in wastewater samples

Read pairs that are not mapped to the reference genome are used for this analysis. Discordant or single-aligned read pairs are not included. Unmapped reads are input to SeqScreen[21] (v4.1, database v23.3, including Refeq archaea, bacteria, eukaryotes and viral sequences) with default parameters. All reads are assigned to a species by SeqScreen with a confidence score. Assignments with confidence score lower than 25% are discarded. The number of reads assigned to SARS-CoV-2 (NCBI Tax ID 2697049), bacteria (NCBI Tax ID 2), eukar-yotes (NCBI Tax ID 2759), viruses (NCBI Tax ID 10239) and archaea (NCBI Tax ID 2157) are counted.

### Reporting summary

Further information on research design is available in the Nature Portfolio Reporting Summary linked to this article.

## Data availability

The sequencing data generated in this study have been deposited in the NCBI SRA database under BioProject PRJNA911448. The 7,629,889 GISAID sequences comprising the MSA and metadata are accessible via the EPI_SET identifier EPI_SET_240619mf [https://doi.org/10.55876/gis8.240619mf]. A full list of GISAID accessions is also available on Figshare https://doi.org/10.6084/m9.figshare.26055322.v1, along with outputs of Variant Database. Sequences and coordinates of primers used in this study are provided in Supplementary Data 3, 4, 10 and 11. All other data supporting the findings described in this manuscript are available in the article and its supplementary files.

## Code availability

Source code, installation guide and usage of Olivar are available on GitHub: https://github.com/treangenlab/Olivar. Olivar primers used in this study can be reproduced with Olivar version 1.1.5[31], available on Zenodo: https://doi.org/10.5281/zenodo.12154937. Input files and instructions for reproducing the Olivar primers can be found on Fig-share: https://doi.org/10.6084/m9.figshare.26055322.v1.

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

## Acknowledgements

We gratefully acknowledge all data contributors, i.e., the Authors and their Originating laboratories responsible for obtaining the specimens, and their Submitting laboratories for generating the genetic sequence and metadata and sharing via the GISAID Initiative, on which this research is based. This work has been supported by CDC contract 75D30122C14709, the Big-Data Private-Cloud Research Cyberinfrastructure MRI-award funded by NSF under grant CNS-1338099, and by Rice University's Center for Research Computing (CRC). B.K. was supported by the NLM Training Program in Biomedical Informatics and Data Science (Grant: T15LM007093). T.J.T. was supported in part by NSF CAREER award IIS-2239114. The authors thank Nina G. Xie for experimental advice, and R. Matt Barnett for his guidance and suggestions on the Olivar web app design and implementation.

## Author contributions

M.X.W., L.B.S. and T.J.T. conceived the project. L.B.S. and T.J.T. supervised the project. M.X.W., N.S., B.K., R.A.L.E., Y.L. and Y.F. developed the software. M.X.W. conducted the software benchmarks. E.G.L and P.K. conducted the sequencing experiments. E.K developed the web application. All authors conceived the experiments and drafted the original paper. All authors read, revised, and approved the manuscript.

## Competing interests

The authors declare no competing interests.
