## [Peer Review File · Nature Communications]

Olivar: automated variant aware primer design for multiplex tiled amplicon sequencing of pathogensREVIEWER COMMENTS

Reviewer #1 (Remarks to the Author):

This manuscript presents *Olivar*, a tool for designing primers for multiplex tiled amplicon sequencing. *Olivar* takes as input a reference sequence, sequence variation information (SNP frequencies) and a blast database of background sequences. It assigns a risk score to every position in the reference genome based on SNP frequencies, GC content, homopolymers and repeated sequence, and based on these risk scores it selects suitable primer design regions (PDRs). These PDRs are then used as input for an existing algorithm, *SADDLE*, which selects primers in these regions such that the chance of dimerization is minimized.

Olivar is presented as an alternative to *PrimalScheme*, the current state-of-the-art primer design tool for multiplex amplicon sequencing. The manuscript includes an *in silico* comparison to primers designed with *PrimalScheme*, which shows that the primers generated by *Olivar* for the SARS-CoV-2 genome score better on several criteria. These *Olivar* primers are also validated experimentally and show slightly better performance than the widely used ARTICv4.1 primers.

The tool comes with a nice web app, I used it on the example data which seemed to work fine. The source code is available on GitLab and is well documented. Installation through conda works and the example runs as expected.

While I believe this manuscript is a valuable contribution to the field and the motivation for developing *Olivar* is clear, I do have some concerns regarding the manuscript as it stands.

* Major comments *

My main concern is the generation of input sequences for *PrimalScheme*. Since *Olivar* takes SNP frequencies as input, while *PrimalScheme* requires genome sequences, the authors construct 100 artificial genomes with the mutations that were input to *Olivar* randomly assigned to genomes. There is no explanation why this is fair. To me it seems that this could highly impact *PrimalScheme*'s performance, since it receives input genomes that do not reflect reality. Have the authors looked into this? What is the impact of different random assignments of SNPs to sequences? And what if they select true SARS-CoV-2 sequences as input for *PrimalScheme*? Moreover, there seems to be a substantial difference (in number) between the primers generated here (for the *in silico* comparison) and the ARTIC primers. Could this be related to the fact that the input genomes used are different from reality?

Table 1 shows an experimental comparison of PrimalScheme and Olivar, but many things remain unclear in the discussion of the results. Which sequence/variant is the synthetic RNA sample? What do the unmapped reads correspond to? Was there any quality filtering?

It is not clear in Table 1 what the coverage means; since values of 0.05 and 0.1 are extremely low, I assume this is normalized by median coverage, as in Figure 5. Why doesn't table 1 include normalized coverage values > 10x?

It is clear that Olivar performs better than ARTIC on the synthetic RNA control, but it is not so evident for the wastewater samples. The main advantage for Olivar seems to be in mapping rates, while coverage appears quite similar. Again, it would be very valuable to know what these unmapped reads correspond to. Is this truly unwanted material, or can it be reads from different lineages that failed to map to the reference genome used? Figure 5 shows coverage results for one of the four wastewater samples taken, which is clearly the best one for Olivar. It would be fair to point this out in the text.

Finally, certain parts of the methods were not entirely clear:

- In Results, "Optimization of PDRs", the authors refer to "a certain level of greediness" but do not explain what they mean.

- The authors also determine a suitable threshold X for considering candidate PDRs, and settle for X=30. Is this threshold specific for SARS-CoV-2, or does this apply more generally?

- In Methods, it is described how the regions C_i are defined. This could use some additional text to further explain each of these steps. In particular, it is not clear where the term "+2l" in C4 and C2k comes from (I assume it is to leave room for another PDR, but this should be explained).

- In "Generation of one PDR", it is not clear how PDRs are selected within the C_i. From Figure 2a I realized that this is a random selection, but the Methods section does not refer to this figure and does not explain it. In general, Figure 2a illustrates the process well and would be helpful in the Methods section.

- Olivar uses SNP frequencies as input, but what about indels? Can these also be taken into account? If not, what does that mean for Olivar in practice?

* Minor comments *

Figure 3 shows an evaluation of Olivar on Nextstrain data using the loss function defined by the authors. They compare to a naive selection of PDRs, but it is not clear what they mean by “naive”. There is a large difference between the loss for the naive approach versus the starting point for Olivar (iteration 0). I’m not sure what the added value is of showing this naive approach, it seems an unfair comparison. If the naive approach is a random selection of PDRs, then it might be a better comparison if the naive approach is also iterative, trying another random selection at every iteration, or to just leave it out.

Results, step 3 “Generation of primer candidates for each PDR and optimization of primer dimers with the SADDLE algorithm”. It would be more specific, hence more clear, to say “minimization” here.

In several places in the manuscript “SARS-CoV-2” is referred to with incorrect typesetting, e.g. Sars-Cov-2.

Methods, Analysis of sequencing data: It would be helpful to add that coverage values are translated into median normalized coverage values.

Methods, calculation of sequence complexity: “where m is the number of a certain k -mer” $\rightarrow m_i$ ($i=0, \dots, n$)

Methods, Primer pool assignment: “the $2k-1$ the PDR pair” \rightarrow the $(2k-1)$ -th PDR pair

The formatting of references needs some work, proper capitalization is missing in many places.

Methods, Calculation of nextstrain nucleotide entropy: “we include global genomes date from” \rightarrow dating from

Reviewer #2 (Remarks to the Author):

This paper developed a new computational tool, Olivar, that enables automated multiplex tiled PCR primer scheme design in a variant (SNP)-aware manner. In general, it is useful to have such tools developed, as multiplex tiled PCR is an effective technique for sequencing pathogens from

matrices with a high level of background genetic material (human samples, wastewater, etc). While I still believe PrimalScheme is still valuable, and that the authors over-stated its shortcomings (e.g. on page 3, end of top paragraph: it is true ARTIC has undergone several iterations, but that would be the same for Olivar if the virus keeps evolving. It is likely that Olivar would need to re-run in the future, once there are new SNPs to feed into it), I also think that Olivar offers some advantages like avoiding primer-dimers and also being 'variant aware'. Therefore, I believe Olivar will be useful to the scientific community.

However, there are a few shortcomings in this paper that need to be resolved before publishing.

1. There were not enough technical replicates performed with the wastewater samples to perform statistical comparisons in the performance of Olivar and PrimalScheme (e.g. Table 1). While it was appreciated that the authors ran paired RNA (originating from same RNA pool) for both schemes on the same samples, there should have been replicate RNA extracts run on the sample dates in order to decipher whether the 'improvements' over PrimalScheme were due to random variation, or were true improvements.

2. The basis for comparison between Olivar and PrimalScheme was not appropriate, given the description of data analysis. Specifically, it was never stated how much data (in bp sequenced) was generated for each sample. Given that genome coverage is a function of the amount of data generated by sequencing, this information is critical. Moreover, for a fair comparison of these datasets, the amount of sequence data should be normalized (e.g. rarified) to the same sampling depth (e.g. same bp of data). This is important in order to be able to correct for any variations in sequencing depth among samples, which is normal within multiplexed runs such as was performed here.

Once the above analyses are added to this paper, the conclusions drawn can be more appropriately supported by the data.

Responses to reviewer feedback are marked in blue and changes to the manuscript are marked in red.

REVIEWER COMMENTS

Reviewer #1 (Remarks to the Author):

This manuscript presents *Olivar*, a tool for designing primers for multiplex tiled amplicon sequencing. *Olivar* takes as input a reference sequence, sequence variation information (SNP frequencies) and a blast database of background sequences. It assigns a risk score to every position in the reference genome based on SNP frequencies, GC content, homopolymers and repeated sequence, and based on these risk scores it selects suitable primer design regions (PDRs). These PDRs are then used as input for an existing algorithm, *SADDLE*, which selects primers in these regions such that the chance of dimerization is minimized.

Olivar is presented as an alternative to *PrimalScheme*, the current state-of-the-art primer design tool for multiplex amplicon sequencing. The manuscript includes an *in silico* comparison to primers designed with *PrimalScheme*, which shows that the primers generated by *Olivar* for the SARS-CoV-2 genome score better on several criteria. These *Olivar* primers are also validated experimentally and show slightly better performance than the widely used *ARTICv4.1* primers.

The tool comes with a nice web app, I used it on the example data which seemed to work fine. The source code is available on GitLab and is well documented. Installation through *conda* works and the example runs as expected.

While I believe this manuscript is a valuable contribution to the field and the motivation for developing *Olivar* is clear, I do have some concerns regarding the manuscript as it stands.

We thank the reviewer for the positive comments and constructive feedback.

* Major comments *

My main concern is the generation of input sequences for *PrimalScheme*. Since *Olivar* takes SNP frequencies as input, while *PrimalScheme* requires genome sequences, the authors construct 100 artificial genomes with the mutations that were input to *Olivar* randomly assigned to genomes. There is no explanation why this is fair. To me it seems that this could highly impact *PrimalScheme*'s performance, since it receives input genomes that do not reflect reality. Have the authors looked into this? What is the impact of different random assignments of SNPs to sequences? And what if they select true SARS-CoV-2 sequences as input for *PrimalScheme*? Moreover, there seems to be a substantial difference (in number) between the primers generated here (for the *in silico* comparison) and the *ARTIC* primers. Could this be related to the fact that the input genomes used are different from reality?

Please allow us to clarify. The reason we constructed 100 artificial genomes is that PrimalScheme limits the number of input genomes to 100. We fetched all Delta (B.1.617.2; 3,961,817 genomes) and Omicron (B.1.1.529; 1,258,730 genomes) genomes from GISAID before Feb. 28, 2022, and thus there were way too many to directly input to PrimalScheme due to the limit.

To work within this 100 genome input limit, we first called SNPs of both Delta and Omicron with Variant Database (v2.4), outputting a list of locations and frequencies of those SNPs on the reference genome (GISAID accession: EPI_ISL_402124). Note that only SNPs with frequency $\geq 1\%$ were kept. Then, we made 100 copies of the reference and added those SNPs back according to their locations and frequencies. Finally, we used these 100 artificially modified genomes as input of PrimalScheme (Methods: SNP calling with Variant Database).

In addition, we set the desired amplicon length to 252~420 for both Olivar and PrimalScheme, instead of 380~420 of the ARTIC primer set. This resulted in more output primers/amplicons than the ARTIC primer set because amplicons are shorter on average. We allowed for a wider range of desired amplicon lengths to increase design flexibility, and to increase the chance of avoiding SNPs and other unwanted features for both Olivar and PrimalScheme. This setting could be more beneficial to Olivar since it optimizes primer placement in parallel, taking more advantage of this increased flexibility. This text is also added to the Results section “Olivar outperforms PrimalScheme in silico” (page 6, line 146-150):

This setting is different from the ARTIC primer set, with amplicon length ranging from 380nt to 420nt. A wider range of desired amplicon length would increase design flexibility, giving more chance of avoiding SNPs and other unwanted features for both Olivar and PrimalScheme. This could be more beneficial to Olivar since it optimizes primer placement in parallel, taking more advantage of this increased flexibility.

As mentioned by the reviewer, random assignment of SNPs might impact the performance of PrimalScheme. We tried 6 different random assignments of SNPs to the 100 artificial genomes, and input to PrimalScheme. Surprisingly, PrimalScheme outputs exactly the same designs for all 6 runs. This is because PrimalScheme always chooses the primer candidate with the least “combined_penalty” (primer.py, class Primer, property combined_penalty; also see region.py, class Region, method _sort_candidates; PrimalScheme source code: <https://github.com/aresti/primalscheme>), consisting of “base_penalty” (intrinsic penalty of the primer, such as GC content) and the sum of “mismatch_penalties”. Primer candidates are all k-mers ($19 \leq k \leq 34$) within a slice of the multiple sequence alignment (MSA). Mismatch penalty is a penalty score of 1 to 3 when a base in the primer does not match a reference in the MSA, with a higher penalty for mismatches close to the 3' end of the primer. Changing the random assignment of SNPs neither changes the least possible base_penalty, nor changes the least possible sum of mismatch_penalties. Thus, random assignment of SNPs does not impact the output of PrimalScheme. This result is noted in “Olivar outperforms PrimalScheme in silico” in Results (page 6, line 143-145):

We tested 6 different random assignments of the 440 SNPs, and PrimalScheme output the same design in all 6 runs, which is expected according to its source code (version 1.4.1).

To further address the reviewer's concern, we compared Olivar and PrimalScheme with a set of real SARS-CoV-2 genomes: 98 variants of interest (VOI) from GISAID. We also set the desired amplicon length to 380~420, leaving less room for Olivar to optimize primer placement. All VOIs are directly input to PrimalScheme. 996 SNPs are called from the MSA of all VOIs, including substitutions, insertions and deletions. Reviewer response Figure 1 shows the frequencies of SNPs overlapping with primers. Note that data points below the gray dashed line are SNPs that occur in only 1 VOI. Since there is randomness in the Olivar optimization process, we ran Olivar 6 times with 6 consecutive random seeds (10 to 15, set by olivar tiling --seed).

Reviewer response Figure 1: Frequency of SNPs in primers, Olivar vs PrimalScheme

With real genomes and more stringent amplicon length requirements, Olivar still outperforms PrimalScheme in terms of avoiding SNPs.

Table 1 shows an experimental comparison of PrimalScheme and Olivar, but many things remain unclear in the discussion of the results. Which sequence/variant is the synthetic RNA sample?

We apologize for the lack of clarity. We have added the following clarification to Methods: Sample preparation and quantification (page 14, line 329-330):

Synthetic RNA control for SARS-CoV-2 is purchased from Twist Bioscience (part number 105204, GISAID accession: EPI_ISL_6841980, GISAID name: Hong Kong/HKU-211129-001/2021, B.1.1.529/BA.1, Omicron variant).

Was there any quality filtering?

After library preparation, all samples are submitted for amplicon sequencing service at Azenta (EZ-Amplicon). Quality filtering and adapter trimming were performed by Azenta. This is mentioned in Methods: Multiplexed PCR, library preparation and sequencing (page 14, line 350-352):

Purified DNA samples were normalized to 20 ng/uL in 25 uL and submitted for amplicon sequencing service at Azenta (EZ-Amplicon), with paired-end (2 × 250bp), quality filtered and adapters trimmed Illumina reads output.

It is not clear in Table 1 what the coverage means; since values of 0.05 and 0.1 are extremely low, I assume this is normalized by median coverage, as in Figure 5. Why doesn't table 1 include normalized coverage values > 10x?

We thank the reviewer for this comment and revised the footnotes for Table 1 to clarify the meaning of coverage. Coverage is the number of reads covering a single nucleotide in the reference genome. Coverage of all nucleotides is compared with the median coverage. Table 1 shows the percentage of nucleotides with less than 0.05× median coverage, or 0.1× to 10× median coverage. We did not include the percentage of nucleotides with more than 10× median coverage because we wanted to focus on the results for low coverage regions. The complete coverage information is now included in the Supplementary Dataset: coverage uniformity.

It is clear that Olivar performs better than ARTIC on the synthetic RNA control, but it is not so evident for the wastewater samples. The main advantage for Olivar seems to be in mapping rates, while coverage appears quite similar.

Coverage for wastewater samples could be worse due to low sample quality (fragmentation, degradation, etc.). However, mapping rate also matters since higher mapping rate reduces sequencing cost for a desired sequencing depth. Olivar has on average 2.3 fold higher mapping rates compared to ARTIC v4.1 (Table 1), which could be beneficial for applications like wastewater pathogen monitoring where a large number of samples need to be sequenced routinely. Olivar also performs comparably to ARTIC on wastewater samples with regards to coverage, and is a fully automated design pipeline.

What do the unmapped reads correspond to? Again, it would be very valuable to know what these unmapped reads correspond to. Is this truly unwanted material, or can it be reads from different lineages that failed to map to the reference genome used?

We thank the reviewer for pointing out the unmapped reads. It is indeed important to find out the source of those reads. Thus, we input them to SeqScreen (v4.1, database v23.3, including Refeq archaea, bacteria, eukaryotes, viral), a sequence screening tool with a sensitive taxonomy classification module. Reviewer response Figure 2 shows the overall result of the taxonomic assignment of all wastewater samples.

Taxonomy assignment of unmapped reads (SeqScreen)

Reviewer response Figure 2: Taxonomy assignment of unmapped reads of all samples (SeqScreen results).

Detailed taxonomy percentages for each sample are shown in the table below.

sample	wastewater site	date	primer	pool	≥25% confidence	SARS-CoV-2	Bacteria	Eukaryotes	Viruses	Archaea	
ART-080822-CB-p1	CB	8/8/22	ARTIC	1	86.3%	0.047%	80.8%	14.1%	1.10%	3.80%	
ART-080822-CB-p2				2	86.8%	0.014%	80.9%	14.5%	1.87%	2.34%	
OLV-080822-CB-p1		8/15/22	Olivar	1	85.2%	0.137%	80.2%	17.7%	0.75%	1.19%	
OLV-080822-CB-p2				2	86.7%	0.046%	73.2%	24.1%	1.38%	1.01%	
ART-081522-CB-p1		KB	8/8/22	ARTIC	1	89.8%	0.253%	72.1%	18.6%	0.73%	8.50%
ART-081522-CB-p2					2	90.4%	0.087%	73.6%	21.8%	0.74%	3.82%
OLV-081522-CB-p1	8/15/22		Olivar	1	85.4%	0.161%	76.4%	20.5%	0.76%	1.86%	
OLV-081522-CB-p2				2	88.1%	0.071%	67.9%	28.6%	1.50%	1.44%	
ART-080822-KB-p1	KB		8/8/22	ARTIC	1	88.5%	0.022%	81.2%	16.4%	0.53%	1.76%
ART-080822-KB-p2					2	82.5%	0.015%	79.8%	17.3%	2.12%	0.64%
OLV-080822-KB-p1		8/15/22	Olivar	1	86.3%	0.048%	77.4%	20.5%	0.86%	1.08%	
OLV-080822-KB-p2				2	86.4%	0.083%	75.1%	21.9%	0.97%	1.81%	
ART-081522-KB-p1		8/15/22	ARTIC	1	89.2%	0.159%	69.1%	26.6%	0.96%	3.22%	
ART-081522-KB-p2				2	91.1%	0.225%	66.9%	26.8%	1.15%	4.53%	
OLV-081522-KB-p1	Olivar		1	90.0%	0.112%	65.3%	31.8%	1.17%	1.51%		
OLV-081522-KB-p2			2	89.3%	0.156%	69.6%	23.7%	1.00%	5.34%		

Reviewer response Table 1: Taxonomy percentage of unmapped reads of all samples (SeqScreen results).

For all 16 wastewater samples, on average 87.6% of unmapped reads are assigned with ≥25% confidence.

Of those reads,

0.102% assigned to SARS-CoV-2 (NCBI Tax ID 2697049),

74.3% assigned to bacteria
21.6% assigned to eukaryotes
2.74% assigned to archaea
1.10% assigned to viruses

The 25% confidence threshold is determined by the confidence distribution of all reads. This threshold is also recommended by SeqScreen developers. Reviewer response Figure 3 shows the distribution of two samples.

Reviewer response Figure 3: Distribution of SeqScreen confidence. ART-080822-CB-p1: wastewater site CB, Aug. 08 2022, ARTIC v4.1 primer pool 1; OLV-080822-CB-p1: wastewater site CB, Aug. 08 2022, Olivar primer pool 1.

For reads with confidence $\geq 25\%$, we plot hierarchical taxonomy assignment with KronaTools (v2.8.1, <https://github.com/marbl/Krona/wiki>). Here, only sample ART-080822-CB-p1 is shown (Reviewer response Figure 4).

Reviewer response Figure 4: Detailed taxonomy assignment of unmapped reads (SeqScreen result of sample ART-080822-CB-p1: wastewater site CB, Aug. 08 2022, ARTIC v4.1 primer pool 1).

To further confirm the taxonomy assignment, we also ran Kraken2 (v2.1.3, Standard database 10/9/2023, including Refeq archaea, bacteria, eukaryotes, viral, plasmid) on the unmapped reads as well (Reviewer response Figure 5, Reviewer response Table 2).

Taxonomy assignment of unmapped reads (Kraken2)

Reviewer response Figure 5: Taxonomy assignment of unmapped reads of all samples (Kraken2 results).

sample	wastewater site	date	primer	pool	classified	SARS-CoV-2	Bacteria	Eukaryotes	Viruses	Archaea	
ART-080822-CB-p1	CB	8/8/22	ARTIC	1	47.1%	0.150%	93.3%	1.73%	3.43%	0.60%	
2				30.5%	0.017%	91.5%	2.73%	3.74%	0.06%		
OLV-080822-CB-p1			Olivar	1	49.5%	0.059%	96.6%	1.87%	0.64%	0.37%	
2				38.0%	0.072%	95.9%	2.44%	0.74%	0.15%		
ART-081522-CB-p1		8/15/22	ARTIC	1	41.7%	0.786%	81.6%	10.87%	1.99%	0.34%	
2				27.5%	0.281%	69.3%	20.90%	4.19%	0.59%		
OLV-081522-CB-p1			Olivar	1	31.2%	0.464%	81.7%	15.68%	1.95%	0.59%	
2				39.0%	0.678%	78.2%	18.05%	3.53%	0.03%		
ART-080822-KB-p1	KB		8/8/22	ARTIC	1	40.9%	0.456%	91.5%	4.69%	1.06%	0.08%
2					28.5%	0.111%	88.6%	8.66%	1.51%	0.26%	
OLV-080822-KB-p1		Olivar		1	35.1%	0.184%	92.0%	5.66%	0.79%	0.77%	
2				31.9%	0.331%	91.9%	5.05%	1.44%	0.61%		
ART-081522-KB-p1		8/15/22	ARTIC	1	40.3%	1.065%	90.8%	4.67%	1.72%	0.20%	
2				29.0%	0.172%	90.9%	6.48%	1.85%	0.18%		
OLV-081522-KB-p1			Olivar	1	32.1%	0.594%	89.3%	3.84%	1.50%	0.97%	
2				34.7%	0.499%	93.2%	3.19%	2.90%	0.23%		

Reviewer response Table 2: Taxonomy percentage of unmapped reads of all samples (Kraken2 results).

We also compared the results of SeqScreen and Kraken2 (Reviewer response Figure 6). Although Kraken2 assigned more reads to SARS-CoV-2, they still only take up less than 1% of all unmapped reads on average.

Reviewer response Figure 6: Comparison between SeqScreen and Kraken2 results. Each data point represents a wastewater sample (data from Reviewer response table 1 & 2). The red cross shows the average percentage.

For reads that are not confidently assigned by SeqScreen ($\leq 25\%$ confidence), or not classified by Kraken2, they might be 1) primer dimers, 2) homologous sequences such as rRNA, and 3) species not included in the database.

From the results above, we have high confidence that more than 99% of the unmapped reads are not from other lineages of SARS-CoV-2.

Figure 5 shows coverage results for one of the four wastewater samples taken, which is clearly the best one for Olivar. It would be fair to point this out in the text.

This is noted in Results: “Sequencing SARS-CoV-2 from wastewater with Olivar primers and ARTIC v4.1” (page 7, line 185-186):

More data about mapping rate and coverage uniformity can be found in Supplementary Dataset. Note that Olivar has the highest coverage for the wastewater sample shown in Figure 4.

Finally, certain parts of the methods were not entirely clear:

- In Results, “Optimization of PDRs”, the authors refer to “a certain level of greediness” but do not explain what they mean.
- The authors also determine a suitable threshold X for considering candidate PDRs, and settle for $X=30$. Is this threshold specific for SARS-CoV-2, or does this apply more generally?

Thank you for this comment. We revised the text to clarify greediness. The threshold X indicates the level of greediness (lower X means more greediness), and $X=30$ applies to designs other than SARS-CoV-2. We updated the text in Results: “Accelerating the PDR optimization process”, under “Optimization of PDRs” (page 4-6, line 108-122):

Intuitively, one could randomly generate a large number of PDR sets and choose the one with the lowest Loss. However, considering the almost infinite combinations of PDR sets, this approach could be inefficient to find the optimal design. To accelerate the optimization process, we introduced a certain level of greediness into the random generation. First, given that PDRs should not overlap with each other, as well as the desired range of amplicon length, a PDR must fall into a certain region of the targeted sequence (Figure 2a). Here, the level of greediness means instead of randomly selecting a PDR within that region, high risk PDRs are excluded from the random selection, with high risk defined as PDR risk greater than X th percentile. We experimented X from 10 to 90, and as X becomes greater, less greediness is introduced (Figure 2bc). Here the reference genome for SARS-CoV-2 is used (GISAID accession: EPI_ISL_402124), with other input data and parameters described in Methods. While Figure 2bc indicates that smaller X is better, there is a higher chance of falling into local optima. Hence, we set X as 30 for better universality. A detailed description of the generation and optimization of PDR sets can be found in the Methods section.

The level of greediness, or the threshold X , is the trade-off between optimization effectiveness and time consumption. When $X=100$, it is guaranteed to find the optimal design, but the time required might be unacceptable. When $X=0$ (always select the best PDR), it degenerates to a sequential process where no randomness is introduced, with each iteration generating the same set of PDRs.

- In Methods, it is described how the regions C_i are defined. This could use some additional text to further explain each of these steps. In particular, it is not clear where the term “+2l” in C4 and C2k comes from (I assume it is to leave room for another PDR, but this should be explained).

We revised the text to clarify the PDR generation process. We added more descriptions and illustrations in “Optimization of PDRs” in Methods (page 11).

The term “+2l” is for reserving space for the next PDR (Reviewer response Figure 6, also included in Methods). If C_4 extends to the left of p_2+2l , there would be no space for the next forward PDR.

Reviewer response Figure 6: If p_4 (blue dashed line) is placed to the left of p_2+2l , there is no room for the next forward PDR.

- In “Generation of one PDR”, it is not clear how PDRs are selected within the C_i . From Figure 2a I realized that this is a random selection, but the Methods section does not refer to this figure and does not explain it. In general, Figure 2a illustrates the process well and would be helpful in the Methods section.

Figure 2a is now referenced in “Generation of one PDR” in Methods, with additional description and illustrations.

- Olivar uses SNP frequencies as input, but what about indels? Can these also taken into account? If not, what does that mean for Olivar in practice?

Olivar supports input of substitutions, insertions, and deletions. Olivar takes a list of locations and frequencies of SNPs as input (columns: ‘START’, ‘STOP’ and ‘FREQ’), which is intuitive for substitutions and deletions (e.g., for a deletion at genome position 100 to 105, just put 100 in the ‘START’ column and 105 in the ‘STOP’ column). This also works for insertions. For example,

for an insertion between genome position 100 and 101, just put 100 in the 'START' column and 101 in the 'STOP' column.

Note that when Variant Database (v2.4) is used for SNP calling, insertions are not called since Variant Database (v2.4) does not support insertion calling.

* Minor comments *

Figure 3 shows an evaluation of Olivar on Nextstrain data using the loss function defined by the authors. They compare to a naive selection of PDRs, but it is not clear what they mean by "naive". There is a large difference between the loss for the naive approach versus the starting point for Olivar (iteration 0). I'm not sure what the added value is of showing this naive approach, it seems an unfair comparison. If the naive approach is a random selection of PDRs, then it might be a better comparison if the naive approach is also iterative, trying another random selection at every iteration, or to just leave it out.

The "Naive" approach means complete random placement of PDRs. Its Loss is higher than Olivar iteration 0 because Olivar does not place PDRs in a completely random way (see Results: "Optimization of PDRs": "Accelerating the PDR optimization process"; also see Figure 2).

We agree with the reviewer that this result does not highlight the effectiveness of Olivar optimization. Instead, we would like to use the Nextstrain entropy as an alternative way to calculate the risk array, and demonstrate that Olivar works with different formulations of risk calculation. We revised the Results: "Olivar designs PDRs that effectively avoid highly variable genomic regions" into Results: "Versatility of the risk array", under "Optimization of PDRs" (page 6, line 123-137):

A risk array can be calculated from a list of SNPs and a BLAST database (Figure 1), or can be defined in other ways according to the application. Here, we used the entropy data from Nextstrain to demonstrate the versatility of the risk array. Nextstrain calculates the Shannon's entropy for each base of the SARS-CoV-2 reference genome (GenBank accession: MN908947.3), based on the multiple sequence alignment (MSA) of genomes available on GISAID. Details about entropy calculation can be found in Methods. We ran the Olivar workflow with the Nextstrain entropy data as the risk array, showing Olivar works with different formulations of risk calculation (Supplementary Figure S1).

Figure 3 was moved to Supplementary Figure S1.

Results, step 3 "Generation of primer candidates for each PDR and optimization of primer dimers with the SADDLE algorithm". It would be more specific, hence more clear, to say "minimization" here.

Changed to minimization.

In several places in the manuscript “SARS-CoV-2” is referred to with incorrect typesetting, e.g. Sars-Cov-2.

Typos are corrected.

Methods, Analysis of sequencing data: It would be helpful to add that coverage values are translated into median normalized coverage values.

Description added.

Methods, calculation of sequence complexity: “where m is the number of a certain k -mer” → m_i ($i=0, \dots, n$)

Corrected.

Methods, Primer pool assignment: “the $2k-1$ the PDR pair” → the $(2k-1)$ -th PDR pair

Corrected.

The formatting of references needs some work, proper capitalization is missing in many places.

Fixed.

Methods, Calculation of nextstrain nucleotide entropy: “we include global genomes date from” → dating from

Corrected.

Reviewer #2 (Remarks to the Author):

This paper developed a new computational tool, Olivar, that enables automated multiplex tiled PCR primer scheme design in a variant (SNP)-aware manner. In general, it is useful to have such tools developed, as multiplex tiled PCR is an effective technique for sequencing pathogens from matrices with a high level of background genetic material (human samples, wastewater, etc). While I still believe PrimalScheme is still valuable, and that the authors over-stated its shortcomings (e.g. on page 3, end of top paragraph: it is true ARTIC has undergone several iterations, but that would be the same for Olivar if the virus keeps evolving. It is likely that Olivar would need to re-run in the future, once there are new SNPs to feed into it), I also think that Olivar offers some advantages like avoiding primer-dimers and also being ‘variant aware’. Therefore, I believe Olivar will be useful to the scientific community.

We thank the reviewer for the kind comments. Changes in the manuscript are marked in blue and red.

Indeed, Olivar needs to be re-run when new variants of a pathogen emerge. The advantage of Olivar is less manual redesign or experimental validation for each output primer set. The ARTIC primer sets are unlikely to be the direct outputs of PrimalScheme, instead they are tweaked by humans based on the original PrimalScheme designs (e.g., to close the gaps between certain amplicons). Moreover, ARTIC was updated from v1 to v3 within months of the outbreak, not because of new variants but primer dimers, which could be avoided during the initial design. Olivar on the other hand, could avoid sequence variations and primer dimers while maintaining full coverage of the genome, reducing the need of redesign or validation.

We changed the language about ARTIC iterations on page 3, line 49-52:

ARTIC has undergone several iterations of manual tweaking and optimization, while updating the primer set is necessary when new variants of a pathogen emerge, issues such as primer dimer and amplicon dropout could be avoided during in silico design (e.g., the primer dimer issue of ARTIC v4.1).

However, there are a few shortcomings in this paper that need to be resolved before publishing.

1. There were not enough technical replicates performed with the wastewater samples to perform statistical comparisons in the performance of Olivar and PrimalScheme (e.g. Table 1). While it was appreciated that the authors ran paired RNA (originating from same RNA pool) for both schemes on the same samples, there should have been replicate RNA extracts run on the sample dates in order to decipher whether the 'improvements' over PrimalScheme were due to random variation, or were true improvements.

Statistical tests were not performed between ARTIC v4.1 and Olivar due to the limited number of samples. However, given we performed a detailed comparison of the performance of both methods that meets or exceeds standards for comparing computational approaches, we are confident the results show that the improvements are not due to random variation. Future experiments on more samples could be performed to demonstrate statistical significance. First, results on synthetic RNA samples show that mapping rates and coverage are consistent across replicates (Table 1). Second, as shown in the "mapping rate" tab of Supplementary Dataset (column "concordant rate"), Olivar has 2.3 fold higher mapping rates on average for all 8 wastewater samples (row number 9-16 and 25-32, 4 wastewater extracts times 2 primer pools). Last but not least, in **Reviewer response Figure 1** Olivar results in fewer SNPs in primers across six replicates.

2. The basis for comparison between Olivar and PrimalScheme was not appropriate, given the description of data analysis. Specifically, it was never stated how much data (in bp sequenced) was generated for each sample. Given that genome coverage is a function of the amount of data generated by sequencing, this information is critical. Moreover, for a fair comparison of these datasets, the amount of sequence data should be normalized (e.g. rarified) to the same

sampling depth (e.g. same bp of data). This is important in order to be able to correct for any variations in sequencing depth among samples, which is normal within multiplexed runs such as was performed here.

We apologize for the lack of clarity in the coverage analysis. All coverage data in this study is already normalized with median coverage. Here, coverage is the number of reads covering a single nucleotide in the reference genome. Coverage normalization is mentioned in the footnote of Table 1 and “Analysis of sequencing data” in Methods.

REVIEWER COMMENTS

Reviewer #1 (Remarks to the Author):

Thank you for addressing my concerns. I am happy to see the extended results, in particular Reviewer Response Figure 1. This experiment is also informative because the dataset of 98 VOIs is highly diverse, hence more challenging. I would thus recommend adding these results to the supplementary material for completeness.

Finally, some typos:

L19: "sequencing of wastewater samples by 6" -- it seems something is missing here?

L26: variant -> variants

L76: dimer -> dimers

L258: is -> are

Reviewer #1 (Remarks on code availability):

I tested the software with the first review of this paper by using it on the example data, which worked fine. The source code is available on github and appears well documented.

I was unable to find which version of the code was used to produce the experiments in the manuscript. For reproducibility, the authors should specify this version and it should be archived.

Reviewer #2 (Remarks to the Author):

Regarding my previous point about a lack of statistics, the authors claim that "we performed a detailed comparison of the performance of both methods that meets or exceeds standards for comparing computational approaches". I would like a reference for such standards for comparing computational approaches that lacks statistics (e.g. what standard was met or exceeded here?). In my opinion, if a new approach is proposed that claims to be superior to the current best-practice (e.g. ARTIC), then it should be compared using enough replicates to perform a statistically informed comparison.

If I interpret the Supplemental Data 1 labels for wastewater samples correctly, in that '080822-CB' (as example) was the same RNA processed with both ARTIC and Olivar, then I performed a paired t-test (two-tailed) on the mapping rates and coverage for the wastewater samples between these two methods. The mapping rate was significantly higher (effect size = 1.5, $p = 0.008$, $n = 8$) for Olivar compared to ARTIC, but the uniformity of coverage (>5) was not significantly different ($p = 0.78$, $n = 4$). This would suggest to me that more reads mapped to the reference genome using Olivar, but that this did not translate to overall better uniformity of coverage across the genome. Therefore, the improvements shown in Table 1 appear misleading, particularly by bolding the uniformity of coverage for Olivar in the wastewater samples as being superior (when likely this was not a significant increase).

This is also problematic as Figure 4 only shows one wastewater sample, and could be skewed to show a 'best case' scenario. For instance, comparing Figures S3, S7, S8, S9 to Figure 4, it appears that more favorable results for Olivar are shown in the main text (Figure 4) for that single sample than other wastewater samples analyzed. I emphasize that the authors should attempt to incorporate the variation across samples as much as possible in their interpretation and presentation of results.

I would appreciate the thoughts of the authors on such a pairwise comparison between the provided mapping rates and coverage, and how it impacts the interpretation of their results.

Responses to reviewer feedback are marked in blue and changes to the manuscript are marked in red.

REVIEWER COMMENTS

Reviewer #1 (Remarks to the Author)

Thank you for addressing my concerns. I am happy to see the extended results, in particular Reviewer Response Figure 1. This experiment is also informative because the dataset of 98 VOIs is highly diverse, hence more challenging. I would thus recommend adding these results to the supplementary material for completeness.

We thank the reviewer for the valuable comments. Results are added to Supplementary Information (Figure S3 and S11).

We also added text in Results: "Olivar outperforms PrimalScheme in silico" (page 6, line 156):

To further compare the performance of Olivar and PrimalScheme on avoiding SNPs, we chose another set of more diverse genomes: 98 variants of interest (VOI) of SARS-CoV-2. 996 SNPs were called from the MSA of all VOIs, including substitutions, insertions and deletions. We set the desired amplicon length to 380~420, leaving less room for Olivar to optimize primer placement. Olivar had fewer SNPs overlapping with primers in all 6 runs, as shown in Figure S3. More details can be found in "In silico comparison of Olivar and PrimalScheme on 98 variants of interest (VOI)" in Methods. A list of the 98 VOIs and the 996 SNPs can be found in Supplementary Dataset.

As well as Results: "Sequencing SARS-CoV-2 from wastewater with Olivar primers and ARTIC v4.1" (page 7, line 190):

To make sure the unmapped reads were truly unwanted material (e.g., not from different lineages of SARS-CoV-2), we performed taxonomic assignment with the highly sensitive SeqScreen (version 4.1) with results shown in Supplementary Figure S11. On average only 0.102% of the unmapped reads were assigned to SARS-CoV-2; thus we conclude those reads were not from other variants of SARS-CoV-2. More details about taxonomy analysis can be found in Methods.

Finally, some typos:

L19: "sequencing of wastewater samples by 6" -- it seems something is missing here?

Changed to "monitoring of SARS-CoV-2 in wastewater".

L26: variant -> variants

L76: dimer -> dimers

L258: is -> are

Typos are fixed.

Reviewer #2 (Remarks to the Author)

Regarding my previous point about a lack of statistics, the authors claim that “we performed a detailed comparison of the performance of both methods that meets or exceeds standards for comparing computational approaches”. I would like a reference for such standards for comparing computational approaches that lack statistics (e.g. what standard was met or exceeded here?). In my opinion, if a new approach is proposed that claims to be superior to the current best-practice (e.g. ARTIC), then it should be compared using enough replicates to perform a statistically informed comparison.

We thank the reviewer for the valuable feedback. We added results of paired t-tests of mapping rates and coverage uniformity. We also revised the text about wastewater experiment results, emphasizing that while we observed significantly higher mapping rates ($p=0.0088$, effect size=1.27), there was no significant difference in coverage uniformity ($p=0.39$, effect size=0.50). Revisions are copied below and were added to Results: “Sequencing SARS-CoV-2 from wastewater with Olivar primers and ARTIC v4.1”, page 7, line 175:

in Houston, USA at two time points. Using the same primer sets and experimental protocol as above, we observed ~~1-to~~ **3-fold significantly** higher mapping rates of Olivar than ARTIC v4.1 ($p=0.0088$, effect size=1.27), shown in Table 1. Olivar also had ~~lower or~~ similar percentage of low coverage bases (~~less than $0.05 \times$ median coverage, $p=0.39$, effect size=0.50~~), compared with ARTIC v4.1 (Table 1). ~~Details about statistical tests can be found in Methods.~~

Figure 4a shows the overall genomic coverage of both Olivar and ARTIC v4.1 for one of the wastewater samples (site: CB, Aug. 15, 2022), with amplicon locations shown in gray lines. ~~Note that Olivar has the highest coverage for this sample.~~ To compare the coverage uniformity of Olivar and ARTIC v4.1, genomic locations in Figure 4a are sorted by coverage (Figure 4b), showing Olivar ~~has fewer bases with low coverage (8.8% vs. 15.6%)~~, and ARTIC v4.1 have 8.8% and 15.6% of low coverage bases, respectively. ~~This is likely due to Olivar designs having shorter amplicon lengths and more overlapping of amplicons since there is a smaller difference for low coverage amplicons (13.7% and 16.2% for Olivar and ARTIC v4.1, respectively), shown in Figure 4c.~~ Coverage of each amplicon is also shown in Figure 4c, with Olivar and ARTIC v4.1 having 13.7% and 16.2% low coverage amplicons, respectively. Coverage of other samples can be found in Supplementary Figure S4~S10. More data about mapping rate and coverage uniformity can be found in Supplementary Dataset. ~~Note that Olivar has the highest coverage for the wastewater sample shown in Figure 4.~~ Details about coverage calculation can be found in “Analysis of sequencing data” Methods.

We also applied changes in Discussion (page 8 line 209):

The improvement in mapping rates highlights that Olivar can provide robust designs capable of being produced at lower sequencing costs. While Olivar and ARTIC v4.1 had similar coverage uniformity on wastewater samples, the ARTIC primer set had undergone several versions of optimization, including concentration tuning of primers, whereas the Olivar primers were automatically designed and pooled in equal concentration, saving significant time and cost that comes with multiple rounds of manual redesign.

Calculation of p values and effect sizes can be found in Methods: Analysis of sequencing data: Statistical tests (page 15 line 384):

For mapping rates, the two primer pools were treated as different samples ($n=8$). For coverage uniformity, pool 1 and pool 2 were combined ($n=4$), and the percentage of genomic positions with

less than 0.05× median coverage was used as a measurement of coverage uniformity. P values were calculated with a paired, two-tailed t-test. Effect sizes were calculated as Cohen's d, with the formula below,

$$d = \left| \frac{\bar{D}}{SD_D} \right|$$

where d is Cohen's d, \bar{D} is the mean of paired difference, and SD_D is the standard deviation of paired difference.

If I interpret the Supplemental Data 1 labels for wastewater samples correctly, in that '080822-CB' (as example) was the same RNA processed with both ARTIC and Olivar, then I performed a paired t-test (two-tailed) on the mapping rates and coverage for the wastewater samples between these two methods. The mapping rate was significantly higher (effect size = 1.5, $p = 0.008$, $n = 8$) for Olivar compared to ARTIC, but the uniformity of coverage (>5) was not significantly different ($p = 0.78$, $n = 4$). This would suggest to me that more reads mapped to the reference genome using Olivar, but that this did not translate to overall better uniformity of coverage across the genome. Therefore, the improvements shown in Table 1 appear misleading, particularly by bolding the uniformity of coverage for Olivar in the wastewater samples as being superior (when likely this was not a significant increase).

We thank the reviewer for performing the statistical tests. Please note that all coverage analyses are based on normalized coverage (original coverage divided by median coverage. Details can be found in Methods: "Analysis of sequencing data"). This normalization makes the Olivar track and the ARTIC v4.1 track intersect at (15,000, 0) in Figure 4b. Without normalization, the Olivar track will be completely above the ARTIC v4.1 track, making them hard to compare. This normalization will not affect the coverage data in Table 1 since it already compares against median coverage (e.g., the percentage of positions with less than 0.05× median coverage). In short, we are comparing the uniformity of coverage between Olivar and ARTIC v4.1, eliminating the influence of the mapping rate. If we compare the absolute coverage, then Olivar would have a huge advantage due to higher mapping rates.

After performing the statistical tests, we conclude that there is no significant difference in coverage uniformity. Please note that we used equal primer concentration when preparing the Olivar primer pools, while the ARTIC v4.1 primer pools were prepared according to the instructions provided by the ARTIC network. This is described in Methods: "Multiplexed PCR, library preparation and sequencing" (page 15, line 360):

ARTIC v4.1 primer panel was purchased from Integrated DNA Technologies (IDT, Artic v4.1 NCOV-2019 Panel, 500rxn, 10011442). Primer pool was prepared by IDT according to the instructions provided by the ARTIC network. Specifically, additional primers were added and certain primers were pooled with double concentration. A copy of the instructions and the link to the original web page can be found in Supplementary Dataset.

Link to the primer pooling instructions:

<https://github.com/artic-network/primer-schemes/tree/master/nCoV-2019/V4.1>

Based on the instructions, certain primers were pooled with double concentration to improve coverage uniformity, while the Olivar primers were fully auto-designed, pooled with equal concentration, and no wet lab optimization. Therefore, we believe the results on wastewater samples still demonstrate the performance of Olivar, though coverage uniformity is similar to ARTIC v4.1.

We also removed the bolding in Table 1.

This is also problematic as Figure 4 only shows one wastewater sample, and could be skewed to show a 'best case' scenario. For instance, comparing Figures S3, S7, S8, S9 to Figure 4, it appears that more favorable results for Olivar are shown in the main text (Figure 4) for that single sample than other wastewater samples analyzed. I emphasize that the authors should attempt to incorporate the variation across samples as much as possible in their interpretation and presentation of results. I would appreciate the thoughts of the authors on such a pairwise comparison between the provided mapping rates and coverage, and how it impacts the interpretation of their results.

We thank the reviewer for pointing this out; we did not intend for Figure 4 to be misleading in any way, only representative of our tools performance. We mentioned that Olivar has the highest coverage for the wastewater sample shown in Figure 4 (Results: "Sequencing SARS-CoV-2 from wastewater with Olivar primers and ARTIC v4.1", page 7, line 187). We moved this sentence upfront to be more clear (page 7, line 179):

Figure 4a shows the overall genomic coverage of both Olivar and ARTIC v4.1 for one of the wastewater samples (site: CB, Aug. 15, 2022), with amplicon locations shown in gray lines. Note that Olivar has the highest coverage for this sample.

We also added this note to the caption of Figure 4 (page 8):

Note that Olivar has the highest coverage for this wastewater sample. Please also refer to coverage figures of other samples in Supplementary Information.

REVIEWERS' COMMENTS

Reviewer #2 (Remarks to the Author):

Thank you for addressing my concerns regarding statistics. I believe the manuscript now more fairly portrays the benefits of Olivar versus ARTIC.

Reviewer #2 (Remarks on code availability):

I was able to install the code via Conda.